# Trait-like visual cortical hyperactivity in trait anxiety

Zhaohan Wu[1,5], Yuqi You[1,2,5], Joshua A. Brown[3], Raymond J. Dolan ®[4] & Wen Li ®[1,3] ✉

Sensory processing varies across individuals, with some traits—particularly sensory hypersensitivity to basic non-valenced stimuli—linked to emotional traits and psychiatric risk. Traditional accounts attribute this sensory–emotion linkage to limbic or prefrontal modulation, but empirical support is limited. Growing evidence suggests sensory cortex itself flexibly encodes value beyond labeled-line analysis. Across four high-density EEG experiments with multi-wave assessments, we identified reliable visual cortical hyperactivity in high trait anxiety. The effect emerged as early as 46 ms, localized to V1/V2, and was specific to the parvocellular pathway. It was reproducible across arousal states, stimulus valence, extended intervals, and paradigms, and evident for both simple (grating) and complex real-world images. Importantly, cortical excitation–inhibition balance (EEG aperiodic exponent/1/f slope) predicted parvocellular responses in low- but not high-anxiety individuals, implicating disrupted E/I modulation. Thus, trait anxiety alters early visual processing, aligning cortical computations with an individual's intrinsic biological propensity from the outset.

Sensory processing varies across individuals, often presenting as stable sensory traits that are also tied to emotional traits[1,2]. Among these, sensory hypersensitivity shows strong associations with anxiety-related traits, such as trait anxiety, behavioral inhibition, and neuroticism, as well as has been linked to clinical disorders[1–5]. Sensory hypersensitivity is particularly well-documented in schizophrenia[6,7] and autism[8]. Recent research, including meta-analyses, indicated that sensory anomalies, especially sensory hypersensitivity, span a spectrum of psychiatric disorders, with anxiety disorders and posttraumatic stress disorder (PTSD) especially implicated[9–13].

Despite the above, we have limited insight into the precise nature of a link between sensory and emotional traits. Examination of its neural basis has focused predominantly on the limbic system, especially the amygdala and hippocampus, and, to a lesser extent, prefrontal regions[14–16]. This aligns with prevailing clinical models, which attribute sensory anomalies in psychiatric disorders to dysfunctions in

prefrontal and limbic regions, driven by their aberrant reentrant projections to the sensory cortex[17–19]. However, empirical support for a limbic-prefrontal origin of sensory traits is sparse and inconclusive[20]. Paradoxically, despite its primary role in sensory processing, the sensory cortex has been largely overlooked in this research, rendering its role in this sensory-emotion linkage poorly understood.

A widespread assumption is that the sensory cortex maintains high fidelity to external stimuli, which implies consistent responses across individuals—unless modulated by higher-order prefrontal or limbic inputs. However, the notion of rigid sensory encoding conflicts with the evolutionary history of the sensory cortex: Phylogenetic evidence indicates that the sensory cortex evolved prior to the emergence of limbic and prefrontal structures, as observed in ancient organisms such as the ancestral amniote[21,22]. Absent these higher-order structures, sensory cortex likely evolved flexible encoding—dynamically shaped by internal states and the biological value of external

[1]Department of Psychology, Florida State University, Tallahassee, FL, USA. [2]Department of Psychology and Behavioral Sciences, Zhejiang University, Hangzhou, China. [3]Louis A. Faillace, MD, Department of Psychiatry and Behavioral Sciences, University of Texas Health Science Center, Houston, TX, USA. [4]Max Planck-University College London Centre for Computational Psychiatry and Ageing Research, University College London, London, UK. [5]These authors contributed equally: Zhaohan Wu, Yuqi You. ✉e-mail: wen.li.1@uth.tmc.edu

stimuli—to support adaptive interactions with the environment. Recently, the idea that the sensory cortex encodes biological value, without higher-order neural input, and adjusts its responses accordingly, has gained traction[23,24]. Importantly, accruing evidence suggests that sensory cortical value encoding correlates with trait anxiety[25,26] while sensory dysfunctions—such as sensory cortical hyperactivity and disinhibition—are implicated in disorders associated with high trait anxiety, including PTSD, schizophrenia, and autism[6,10,23,27–30].

We hypothesized that sensory hypersensitivity in trait anxiety involves a significant, active contribution from the early sensory cortex. Mechanistically, the sensory cortex operates through well-organized local circuits with rich feedforward and feedback excitation and inhibition, where the excitation-inhibition (E/I) ratio regulates sensory cortical activation levels[24,31]. Indeed, E/I imbalance has been associated with psychiatric disorders, especially those involving significant sensory anomalies, such as schizophrenia and autism[32]. On this basis, we also hypothesized that disrupted E/I modulation in trait anxiety contributes to sensory cortical hyperactivity.

To test these hypotheses, we characterized trait-like sensory cortical hyperactivity and examined its association with trait anxiety through four high-density Electroencephalogram (hdEEG) experiments. To ascertain the reliability and generalizability of a sensory trait and its relation with trait anxiety, we included repeated assessments (up to three waves) and three independent subject samples. To address potential confounds, we implemented aversive conditioning between assessments (in *Experiment 1*) in order to evoke state anxiety and arousal and, in so doing, tease out potential effects of state (vs. trait) anxiety in heightening sensory cortical processing[33,34]. Early visual cortical activity was assayed using visual evoked potentials (VEPs), an electrophysiological feature emerging from the early (primary and secondary) visual cortex, at approximately 50–120 ms post stimulus onset[35,36]. In generating characteristic VEPs, visual processing in the primate brain is mediated by distinct pathways, especially the parvo- (P) and magno-cellular (M) visual pathways[35–37]. These pathways serve specialized functions[37] and are differentially implicated in psychiatric disorders[6,38,39]. To precisely delineate their contribution to sensory hyperactivity, we employed carefully calibrated M- and P-selective stimuli across all four experiments.

## Results

### Stable and trait-like visual cortical biases in trait anxiety
In *Experiment 1*, we applied two sets of stimuli known to selectively elicit M and P-pathway processing—M-pathway: achromatic, low-spatial-frequency, and low-contrast Gabor patches; P-pathway: chromatic (red/green), high-spatial-frequency, and individually-determined isoluminance Gabor patches[37]. Participants ($N = 52$; mean age, 19.5 years; 22 men) performed a simple orientation discrimination task during high-density (HD; 96-channel) EEG recordings[25] on Day 1 and Day 16 (Fig. 1).

To examine stable and trait-like sensory cortical biases, we administered the task at three distinct time points: Time 1: baseline; Time 2: after a standard aversive conditioning manipulation; and Time 3: Day 16 (15 days after conditioning). Consistency in response between Time 1 and Time 2 would support trait-like reliability that persists across different arousal and anxiety states, while consistency with Time 3 would attest to robust test-retest reliability. At all three time points, trials for the conditioned and non-conditioned stimuli (CS and non-CS) were pooled. Additionally, we also analyzed CS and non-CS trials separately, which yielded virtually identical results (see details in SOM and Table S1). Trait anxiety was assessed with the Behavioral Inhibition Scale (BIS)[40], a neurobiologically motivated questionnaire with high reliability and high predictive validity for anxiety[41,42]. Importantly, the trait of behavioral inhibition is closely related to sensory sensitivity[43].

As established in the literature[35,36], the M- and the P-selective Gabor patches evoked a positive-going (P1) component and a negative-going

(C1/N1) complex, respectively (Fig. 1B–D). These VEPs were extracted from the Oz site (collapsed across 3 midline electrodes), where they were maximally distributed, and were submitted to Pearson correlational analyses. We observed a positive correlation between C1/N1 magnitude (i.e., inverted amplitude to adjust for its negative polarity) and trait anxiety at all three time points (T1: $r = 0.37$, $p = 0.011$; T2: $r = 0.38$, $p = 0.009$; T3: $r = 0.37$, $p = 0.026$), suggesting parvocellular visual cortical hyperactivity in trait anxiety. We also found a negative correlation between P1 amplitude and trait anxiety at all three time points (T1: $r = -0.35$, $p = 0.015$; T2: $r = -0.38$, $p = 0.008$; T3: $r = -0.33$, $p = 0.050$), suggesting magnocellular visual cortical hypoactivity in trait anxiety. Finally, leveraging the hdEEG recordings, we also performed intracranial source-level analyses on the C1 and P1 components using exact low-resolution electromagnetic tomography (eLORETA)[44], which confirmed their associations with trait anxiety in early visual cortex (V1/V2; see SOM and Fig. S2).

In addition, at all three time points, point-by-point analysis of the waveforms throughout the early visual processing stage (i.e., 0–200 ms post-stimulus) confirmed continuous intervals encompassing the P- and M-evoked VEPs (gray boxes in Fig. 1), where the above correlation was reliably significant after correction for multiple tests. Specifically, for the P-evoked waveforms, the continuous intervals were 86–175 ms; 86–171 ms; and 86–160 ms for Time 1, Time 2, and Time 3, respectively (see details in SOM). For the M-evoked waveforms, the continuous intervals were 72–200 ms, 42–200 ms, and 78–200 ms for Time 1, Time 2, and Time 3, respectively. Finally, using Hittner's Monte Carlo evaluation method[45], we compared the correlation across the three time points and observed no effects of time ($p$'s > 0.44; see details in SOM), indicating the reliability and robustness of this association across different arousal levels and across extended time. Thus, these findings are consistent with the presence of stable, trait-like, visual cortical biases in trait anxiety.

An opposing pattern of M- and P-biases in trait anxiety was not predicted. Intriguingly, there were strong negative correlations between the magnitudes of VEPs for the two respective pathways, $r$'s < $-0.62$, $p$'s < 0.001 (see SOM Fig. S1). These negative correlations rule out alternative accounts of general differences in neural responsivity or sensory reactivity across individuals (as they would predict positive correlations). In addition, the finding might indicate a reciprocity between parvo- and magno-cellular functioning within an individual. However, given that M- and P-selective trials were intermixed and presented in rapid succession, it is also possible that a repetition-suppression type of effect between the two pathways might drive this reciprocity. In addition, an alternative explanation for the parvocellular hyperactivity is that it reflects a restricted sensitivity for color processing. Based on these considerations and to adjudicate between the aforementioned explanations, we conducted *Experiments 2 and 3*.

### Reproducible and stable association of trait anxiety with parvocellular visual cortical hyperactivity
*Experiments 2 and 3* (order counterbalanced across subjects) were conducted in an independent subject sample ($N = 52$; mean age, 20.0 years; 16 men). To minimize potential effects of repetition suppression between the two pathways, *Experiment 2* presented the P- and M-selective Gabor patches in non-overlapping visual fields, thereby activating separate populations of retinal and visual cortical cells. Specifically, based on their relative sensitivity to foveal versus peripheral stimuli[37], we presented P- and M-selective Gabor patches at center and periphery (at 10° eccentricity), respectively (Fig. 2A). Similar methods (e.g., EEG recordings and preprocessing, trait anxiety measurement, statistical analysis) as in *Experiment 1* were applied here and in the other experiments unless otherwise stated.

The P-selective stimuli evoked a C1/N1 complex at Oz (Fig. 2B, bottom). The peripherally presented M-selective stimuli

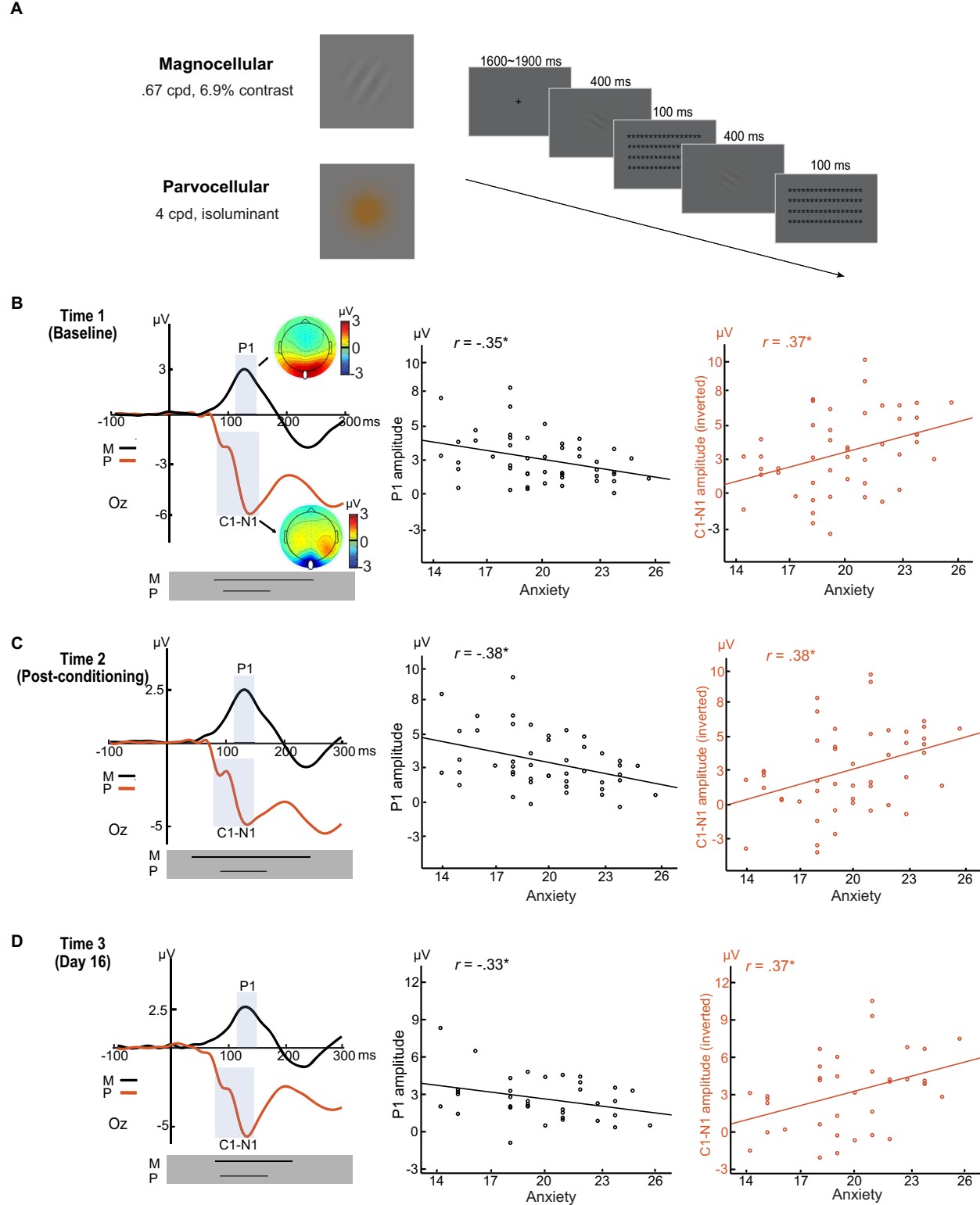

evoked a clear (albeit attenuated) P1 at the contralateral posterior sites (P7/P8; Fig. 2B, top). Importantly, we replicated a positive correlation between trait anxiety and C1 magnitude ($r = 0.33$, $p = 0.018$). In addition, source-level analysis in eLORETA further confirmed the association in early visual cortex (V1/V2; see SOM and Fig. S3, left). A point-by-point analysis in the 200-ms post-stimulus window confirmed a continuous interval (58–105 ms; encompassing the C1 potential) where the correlation was reliably significant after correction for multiple tests (see details in SOM). However, a negative correlation between trait anxiety and P1 evoked by M-selective stimuli observed in *Experiment 1* was not replicated here ($r = 0.03$, $p = 0.81$), while point-by-point analysis in the 200 ms window failed to isolate any time points showing a correlation with trait anxiety ($p > 0.10$). These findings are again supportive of a selective association of trait anxiety with parvocellular visual cortical hyperactivity (but not with magnocellular activity).

*Experiment 3* was designed to exclude the alternative explanation of a restricted sensitivity for color processing and to ascertain the

**Fig. 1 | Stable and trait-like visual cortical biases in trait anxiety (Experiment 1).**
**A** *Example stimuli and trial*. The M-selective stimulus (Top) comprised a Gabor patch that contained a gray grating with a low spatial frequency of 0.67 cycles per degree (cpd) and low contrast of 6.9% (Michelson contrast). The P-selective stimulus (Bottom) comprised a Gabor patch that contained a green/red grating with high spatial frequency of 4 cpd. Notably, the green/red grating and the background were isoluminant (based on individual heterochromatic flicker photometry) such that magnocellular activation induced by the onset of these stimuli would be maximally inhibited. Participants performed an orientation discrimination task, which contained trials of two consecutively delivered Gabor patches of either the Same or Different orientation. M- and P-selective trials were randomly intermixed. *VEPs and their correlation with trait anxiety at Time 1* (**B**), *Time 2* (**C**), and *Time 3* (**D**). The orientation discrimination task was administered at three time points,

including Time 1—the baseline, Time 2—immediately after aversive conditioning, and Time 3—15 days later. The M- and P-selective stimuli evoked characteristic P1 (black waveforms) and C1/N1 (red waveforms) potentials, respectively, both maximally distributed around Oz (indicated by white dots in the topographic maps). In the scatter plots, the C1-N1 amplitude was inverted (as magnitude) to adjust for its negative polarity. Time 1: Parvocellular $r = 0.37$, $p = 0.011$; Magnocellular $r = -0.35$, $p = 0.015$; Time 2: Parvocellular $r = 0.38$, $p = 0.009$; Magnocellular $r = -0.38$, $p = 0.008$; Time 3: Parvocellular $r = 0.37$, $p = 0.026$; Magnocellular $r = -0.33$, $p = 0.050$; * $p < 0.05$ two tailed. Black lines (in the gray bars beneath the VEPs) = windows of continuous significant correlation. Notably, these lines fully encompassed the VEP windows predefined based on ERP conventions, providing empirical validation for their use. Results were based on Pearson correlation analysis, and $p$-values were two-tailed. Source data are provided as a Source Data file.

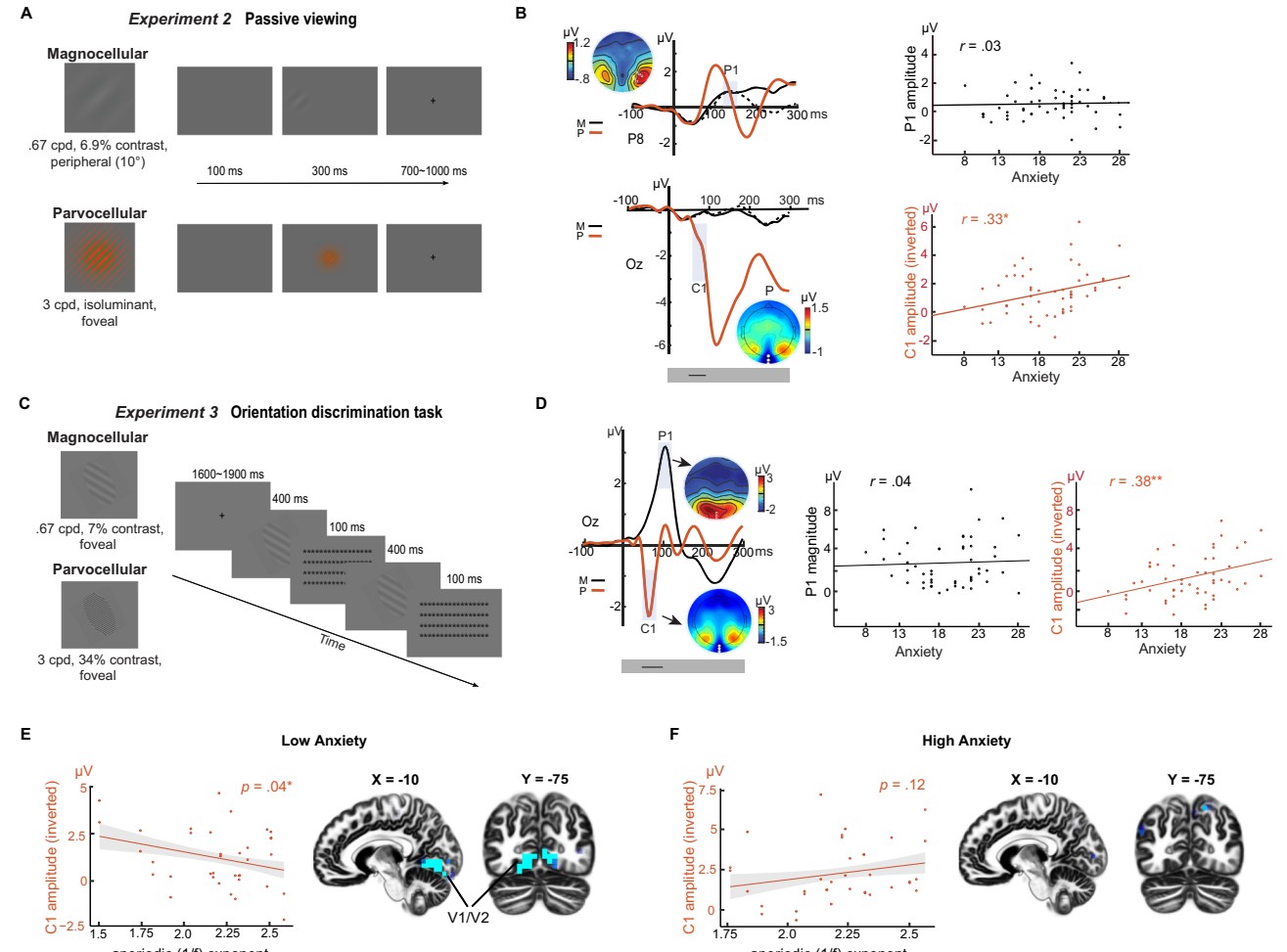

**Fig. 2 | Reliable parvocellular visual cortical hyperactivity in trait anxiety (Experiments 2 and 3). A** *Experiment 2 design*. Participants passively viewed M-selective (Top; low spatial frequency/contrast, 10 degrees eccentricity) and P-selective (Bottom; high SF, red/green isoluminant, fovea) stimuli. **B** *Experiment 2 results*. Top row: M stimuli evoked a P1 potential maximal at P7/P8 (white dots in topographical map; solid black waveform = M stimuli in contralateral hemifield; dotted black waveform = M stimuli in ipsilateral hemifield). Bottom row: P stimuli evoked a C1/N1 complex (red waveform) maximal at Oz (white dots in topographical map), which correlated with trait anxiety. Parvocellular $r = 0.33$, $p = 0.018$; Magnocellular $r = 0.03$, $p = 0.81$. **C** *Experiment 3 design*. Participants performed an orientation discrimination task with gray M-selective (Top; low spatial frequency/contrast) and P-selective (Bottom; high spatial frequency/contrast) Gabor patches. Stimuli were randomly intermixed and centrally presented.

**D** *Experiment 3 results*. M stimuli evoked a P1 maximal at Oz. P stimuli evoked a C1 maximal at Oz, which correlated with trait anxiety. Parvocellular $r = 0.38$, $p = 0.006$; Magnocellular $r = 0.04$, $p = 0.760$. **E, F** *Disrupted* E-I *modulation of parvocellular processing in trait anxiety*. Resting-state EEG aperiodic exponent (1/f slope) indexed E–I balance (flatter slope = excitation/disinhibition bias). In low trait-anxiety participants, excitation bias predicted larger C1 amplitudes (source localized to V1/V2; display threshold: $p < 0.005$ uncorrected; **E** whereas no association was observed in high trait-anxiety participants (**F**). Scatterplots display C1 amplitude as magnitude (inverted for negative polarity). Black lines (in gray bars beneath the VEPs) = windows of continuous significant correlation; shaded areas = 95% CI. * = $p < 0.05$ two-tailed; ** = $p < 0.01$ two-tailed. Results were based on Pearson correlation analysis (**B, D**) and mixed regression analysis (**E, F**), and $p$ values were two-tailed. Source data are provided as a Source Data file.

retest reliability of this effect. Specifically, achromatic P-selective stimuli were used with high spatial frequency and high contrast retained to preferentially activate the parvocellular pathway (Fig. 2C). In support of the stimulus manipulation, these stimuli evoked a characteristic C1 potential (Fig. 2D). Importantly, the C1 again exhibited a positive correlation with trait anxiety ($r = 0.38$, $p = 0.006$). In addition, source-level analysis in eLORETA further confirmed an association within early visual cortex (V2; see SOM and Fig. S3, right). Similar to *Experiment 1*, a point-by-point analysis throughout the 200 ms post-stimulus window again revealed a continuous interval (46–94 ms; encompassing the C1 potential) where the correlation remained reliably significant after correction for multiple tests (see details in SOM). The M-selective stimuli were centrally presented and evoked a strong P1 potential at Oz. As in *Experiment 2*, we did not observe a significant correlation between trait anxiety and P1 ($r = 0.04$, $p = 0.76$) or any time point in the 200 ms window ($p$'s > 0.10). Finally, we found no difference in the correlation between the two experiments ($p$'s > 0.64), highlighting a consistency in association, despite changes in both visual properties and tasks. Together, the results of *Experiments 2 and 3* confirm a reliable and replicable parvocellular visual cortical hyperactivity in trait anxiety, while failing to support the presence of a magnocellular hypoactivity in trait anxiety (see SOM for Bayesian Factors in support of the null finding).

### Disrupted E-I modulation of parvocellular visual cortical activity in trait anxiety

As discussed above, sensory cortical functioning is regulated by cortical E/I balance, and disruptions in this balance (i.e., E/I imbalance) are implicated in various psychiatric disorders, especially those involving significant sensory anomalies. Thus, we hypothesized that disrupted E/I modulation in high trait anxiety serves as a mechanism underlying its associated sensory cortical hyperactivity. To test this, we recorded resting-state EEG among participants in *Experiments 2 and 3* and extracted an index of E/I balance—the aperiodic exponent (1/f slope)—from the EEG power spectrum (3–50 Hz). Converging evidence, variously derived from computational modeling, intracranial recordings, and EEG data, suggests that the 1/f slope is sensitive to excitatory and inhibitory interplay at the level of the synapse, as well as GABAergic pharmacological modulation[46–50]. Importantly, the 1/f slope closely tracks E/I ratios: flatter slopes reflect higher E/I ratios (greater excitation), steeper slopes lower E/I ratios (greater inhibition)[49,51].

Therefore, we submitted C1 magnitudes from both *Experiments 2 and 3* into a mixed-effects regression model with trait anxiety, experiment, aperiodic 1/f slope, and their interaction entered as regressors. In support of our hypothesis, we observed a significant interaction between the 1/f slope and trait anxiety in predicting C1 magnitude (coefficient = 0.46, $SE = 0.20$, $p < 0.05$). To further probe this interaction, we performed a median split based on trait anxiety level and applied a mixed-effects regression model on the 1/f slope and C1 magnitudes in the high- and low-anxiety groups separately (Fig. 2E, F). In the low-anxiety group, we confirmed a negative association between C1 magnitude and 1/f slope ($p = 0.04$ one-tailed), suggesting that, in low anxiety, stronger parvocellular visual cortical response is linked to higher E/I ratios (Fig. 2E). Our eLORETA source-level analyses on the C1 component further localized this association between C1 magnitude and E/I ratio to early (primary and secondary) visual cortex (V1/V2; peak $x = -10$, $y = -75$, $z = -10$, $r = -0.53$, $k = 100$; FDR $p < 0.05$; Fig. 2E). By contrast, in the high-anxiety group, no significant association between 1/f slope and C1 magnitude emerged from either the surface- ($p = 0.12$ one-tailed; and potentially trending in the opposite direction) or source-level analyses (Fig. 2F). Together, results here suggest that E/I modulation of the visual cortex was somewhat compromised in high trait anxiety, potentially resulting in visual cortical hyperactivity among such individuals.

### Ecological validity: parvocellular visual cortical hyperactivity to real-world images in trait anxiety

The above experiments used basic laboratory stimuli and, arguably, do not address ecological validity and real-world relevance of the observed association. A more general question is whether trait-anxious individuals exhibit visual cortical hyperactivity while navigating their natural environment. To address this, we examined visual cortical responses to natural scene images in association with trait anxiety. Specifically, we reanalyzed data from a prior study ($N = 46$; 19.3 years; 22 men) using images conveying complex natural scenes[52]. Critically, the images were filtered to contain high or low spatial frequency (HSF/LSF) that selectively activate the parvocellular or magnocellular pathway, respectively (Fig. 3A). Of note, these images were carefully calibrated to ensure that P- and M-selective images were comparable in luminance, contrast, wavelength energy, and visual complexity (edge density, entropy, compressed image size)[52]. Finally, the images contained neutral, disgust, and fear emotions, allowing us to also examine the effect of emotion on the observed association.

Both HSF and LSF stimuli evoked a P1 component. Amplitude of the P1 evoked by P-selective (HSF) images again correlated positively with trait anxiety ($r = 0.38$, $p = 0.017$), replicating the above findings with Gabor patches (Fig. 3B). In addition, source-level analysis in eLORETA further confirmed an association within early visual cortex (V1/V2; see SOM and Fig. S4). Moreover, a point-by-point analysis of the 200 ms post-stimulus window again highlighted a continuous interval (101–140 ms; encompassing the P1 potential) wherein the correlation was reliably significant after correction for multiple tests (see details in SOM). By contrast, the amplitude of the P1 evoked by M-selective (LSF) images did not correlate with trait anxiety ($r = 0.15$, $p = 0.35$), while a point-by-point analysis also failed to identify any time points showing a significant correlation ($p$'s > 0.12). Finally, consistent with findings from *Experiment 1*, we found no effect of emotion on the correlation for both P- and M-selective images (P-selective: $p$'s > 0.89; M-selective: $p$'s > 0.54), and the correlation coefficients were highly comparable across the three emotional categories (P-selective: $r = 0.36$–$0.37$; M-selective: $r = 0.08$–$0.21$). Thus, these results again confirmed an association between trait anxiety and parvocellular visual cortical hyperactivity, while validating the ecological validity of this association.

To ensure rigor and robustness of the findings, we further submitted the six $p$-values from the six correlational tests between trait anxiety and P-selective VEPs, across *Experiments 1–4*, to Benjamin-Hochberg correction for multiple comparisons. All six remained significant at the adjusted thresholds (see SOM).

### Sex independence: Similar trait anxiety association with parvocellular visual cortical hyperactivity in women and men

There is a well-established sex difference in prevalence and clinical manifestations of anxiety[53], potentially reflecting distinct pathophysiological processes. On this basis, in hierarchical multiple linear regressions on pooled data from our three independent samples (see SOM for details), we explored whether women and men differ with respect to an association between trait anxiety and visual cortical hyperactivity. Separate models were implemented for P- and M-selective VEPs, with Study Sample (dummy-coded), Sex, and Trait Anxiety (BIS scores) entered in the first step and Sex-by-Anxiety in the second step as regressors. For the P-selective VEP, we replicated the significant anxiety effect ($sr = 0.35$, $p < 0.001$) but observed no Sex-by-Anxiety interaction ($sr = -0.03$, $p = 0.762$). For the M-selective VEP, there was no effect of anxiety ($sr = -0.02$, $p = 0.864$), consistent with the main results, and no sex-by-anxiety interaction ($sr = -0.004$, $p = 0.960$).

We also performed similar regression analyses in female and male groups separately. In both groups, trait anxiety significantly predicted P-selective VEPs (females/males: $sr = 0.36/0.35$; $p = 0.001/0.011$). By

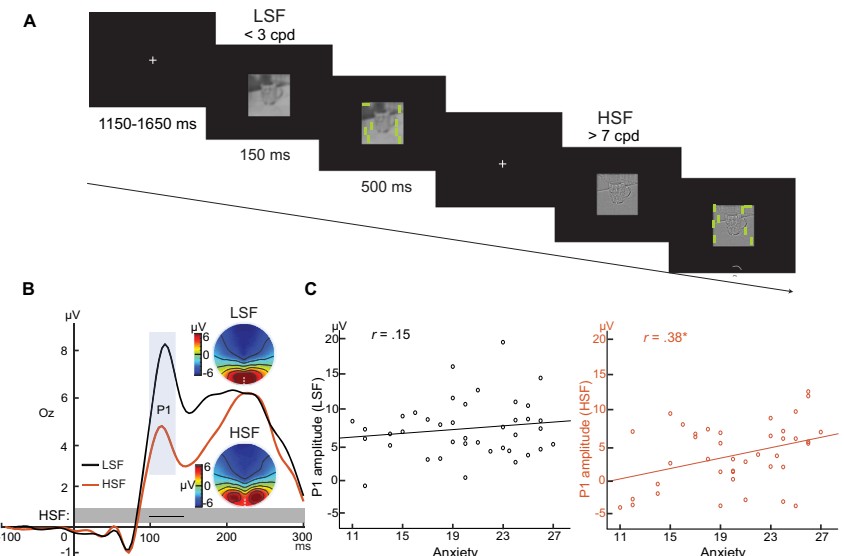

**Fig. 3 | Ecological validity: Parvocellular visual cortical hyperactivity in trait anxiety applicable to natural scenes (Experiment 4). A** *Example stimuli and trials in Experiment 2*. The M- and P-selective stimuli were gray scale with LSF (<3 cpd) and HSF (>7 cpd), respectively. A visual search array containing a horizontal bar among vertical bars appeared after 150 ms, and participants were asked to indicate the quadrant containing the horizontal bar. **B**, **C** VEPs and their correlation with trait anxiety. **B** Both stimuli evoked P1 potentials maximally distributed around Oz (white dots in the topographic maps). **C** Significant association with trait anxiety was observed for the P1 evoked by P-selective stimuli only. Parvocellular $r = 0.38$, $p = 0.017$; Magnocellular $r = 0.15$, $p = 0.352$. Black lines (in gray bars beneath the VEPs) = windows of continuous significant correlation. * = $p < 0.05$ two-tailed. Results were based on Pearson correlation analysis, and $p$-values were two-tailed. Source data are provided as a Source Data file.

contrast, neither group showed a significant association between trait anxiety and M-selective VEPs (females/males: $sr = -0.03/-0.01$; $p = 0.760/0.948$). Notably, a simple t-test on BIS scores showed significantly greater trait anxiety in females ($20.27 \pm 4.23$) compared to males ($18.68 \pm 4.08$), $t_{136} = 2.23$, $p = 0.027$, consistent with the known sex bias in anxiety[53] and suggesting also an absence of systematic sampling bias in our study. In summary, we found no evidence that sex moderates the relationship between trait anxiety and visual cortical hyperactivity, suggesting that this sensory cortical hyperactivity in trait anxiety is independent of sex.

## Discussion

Across four experiments and three independent subject samples, we show that early visual responses (VEPs) consistently manifest a reliable and generalizable pattern of parvocellular visual cortical hyperactivity linked to the presence of trait anxiety. The trait-like nature of this hyperactivity was reproducible across varying arousal states, neutral and negative stimuli, extended intervals, and diverse experimental paradigms. We also demonstrate ecological validity and real-world relevance to these findings using data involving the processing of complex natural scenes, beyond simple laboratory stimuli. Notably, while female participants showed higher levels of trait anxiety than male participants, they exhibited a comparable association between trait anxiety and visual cortical hyperactivity, suggesting sex independence of this effect. Mechanistically, a marker of E/I ratio predicted parvocellular VEPs in low (but not high) trait anxiety, suggesting a linkage between visual cortex hyperactivity in trait anxiety and E/I modulation. Therefore, sensory encoding of even basic, non-valenced stimuli can be biased in trait anxiety. Such biases could act as a bottom-up mechanism, shaping downstream, higher-order processing and initiating a cascade of sensory, affective, and cognitive symptoms characteristic of anxiety and related disorders.

Previous work has considered sensory biases as state-dependent (e.g., linked to state anxiety)[33,34], but the current finding challenges this assumption. To the extent that arousal and state anxiety heighten sensory processing[33,34], our experimental manipulation of arousal and anxiety states via aversive conditioning (*Experiment 1*) indicated an equivalent relationship between trait anxiety and VEPs before and after conditioning, effectively ruling out arousal or state anxiety as explanations for our findings. Trait anxiety is known to upmodulate early sensory processing of threat cues[54–56]. By including both neural and emotional stimuli (i.e., aversively-conditioned Gabor patches in *Experiment 1* or scenes depicting disgust or fear in *Experiment 4*), we show comparable effects, indicating that this sensory trait and its association with trait anxiety are independent of emotion.

We acknowledge that non-invasive approaches, including stimulus manipulation, cannot achieve a complete visual pathway separation. Nonetheless, our integrated calibration of stimulus luminance, contrast, color, and spatial frequency provides for a maximization of pathway dissociation with consistent elicitation of VEPs characteristic of each pathway. Notably, although *Experiment 1* showed magnocellular hypoactivity in trait anxiety, this effect was absent in *Experiments 2 and 3*, where task design (particularly, the parafoveal magnocellular presentation in *Experiment 2*) minimized suppression of magnocellular responses from rapid parvocellular stimulus repetition. Importantly, these results contrast with reproducible parvocellular hyperactivity in trait anxiety, helping exclude broader confounds such as general biases in attention or neural reactivity associated with trait anxiety.

While previous personality research has linked sensory traits to emotional traits[1–5], these studies focused predominantly on limbic and prefrontal regions as putative sites of origin, with inconclusive empirical support[20]. They have also emphasized a magnocellular primacy, wherein a fast magnocellular input to limbic and prefrontal areas is assumed to drive sensory cortical biases via reentrant feedback[17–19]. This "magnocellular advantage" has been incorporated into the theoretical conceptualization of diverse psychiatric disorders, wherein biased magnocellular processing contributes to a range of cognitive and affective symptoms[57].

Strikingly, our experiments consistently highlight parvocellular (but not magnocellular) cortical hyperactivity in trait anxiety, contradicting a canonical magnocellular-dominant, top-down account. Nonetheless, our findings align with evidence from autism research,

which suggests an enhanced parvocellular (vs. magnocellular) hyperactivity in this disorder[39,58]. It should be noted that the parvocellular pathway operates at a slower speed compared to the magnocellular pathway, reaching the visual cortex (V1) at ~50 ms. Consistent with this, we detected parvocellular visual cortical hyperactivity in trait anxiety as early as 46 ms (Fig. 2D) and confirmed its emergence in the course of an initial feedforward sweep. In this regard, our study supports a "bottom-up" view wherein a sensory-emotion connection can originate in the early sensory cortex, biasing early sensory processing and influencing subsequent processes.

Our findings motivate a revised conceptualization of key features seen in disorders such as autism, anxiety, and PTSD. Anatomically, the parvocellular pathway is the dominant component of the visual system, constituting about 80% of the ganglion cell population and providing the primary input to the visual cortex[37,57]. As the gateway between the brain and the external world, the sensory cortex plays a pivotal role in gating and filtering sensory input[59]. In individuals with high trait anxiety, sensory cortex hyperactivity (and disinhibition) might facilitate processing of irrelevant, unwanted, or subthreshold environmental cues, particularly those falling on the fovea, to evoke full-blown sensory cortical responses. The ensuing excess of sensory output, or "neural noise," could in turn overwhelm downstream, high-order brain areas, triggering a cascade of multi-faceted disruptions that manifest in the diversity of symptoms seen in PTSD, autism, and related disorders[29,60,61].

This general hyperactivity can further interact with another key process in sensory cortex: sensory cortex is recognized as a critical storage site for threat memory[24,62,63], especially in individuals with high trait anxiety[25,26]. Consequent to visual cortical hyperactivity, threat memory representations stored in visual cortex could be more likely to be reactivated, resulting in heightened threat memory recall. This type of interaction could, in turn, contribute to intrusive threat or trauma memories commonly seen in anxiety, PTSD, and autism[42,43,64]. Furthermore, the parvocellular visual pathway plays a central role in detailed, fine visual analysis[37,57], and hyperactivity might contribute to excessive fine-detail processing reported in anxiety and autism[65,66]. This bias could conceivably extend to heightened detection of minute and insignificant threat cues, a defining feature of anxiety[55,67].

Sensory cortical activity is regulated at multiple levels. At a fine-grained microscopic level, it is influenced by excitatory and inhibitory inputs within local circuitry[24,31]. This E/I balance (or the lack thereof) is believed to play a fundamental role in psychiatric disorders[32]. Specifically, sensory hypersensitivity (e.g., "superior low-level perceptual processing"), a hallmark symptom of autism[68], is linked to deficient E/I regulation of the sensory cortex[60]. Our findings related to E/I balance in the visual cortex suggest a similar mechanism may be at play in trait anxiety. Thus, in low trait anxiety, an E/I ratio predicted the magnitude of parvocellular VEPs, demonstrating appropriate excitatory and inhibitory modulation of visual cortical activity. In contrast, this was not the case in high trait anxiety, suggesting a disruption of E/I modulation might drive visual cortical hyperactivity in these individuals.

While GABAergic (inhibitory) and glutamatergic (excitatory) activity in local circuits are thought to directly maintain an E/I balance[31], midbrain and hindbrain monoaminergic afferents are also considered to play a role. Although GABAergic and glutamatergic anomalies in trait anxiety are not yet well specified, monoaminergic dysfunctions are widely reported[60,69]. These monoaminergic imbalances, such as serotonergic deficiency, may contribute to the sensory-anxiety association by disrupting E/I regulation of the sensory cortex, thereby exaggerating sensory cortical activity[43]. Conversely, anxiolytic drugs, including GABAergic treatments (such as diazepam and gabapentin) and monoaminergic treatments (such as serotonin selective receptor inhibitors), could conceivably alleviate anxiety by downregulating sensory cortical activity through enhancing cortical inhibition.

Several limitations warrant discussion. Trait research, by emphasizing stable dispositions, limits experimental manipulation and largely relies on correlational analyses. While interventional techniques, such as pharmacological manipulation and neurostimulation, can modulate anxiety or sensory processing, they induce transient, state-level changes that may confound trait-level associations. Establishing causality will require longitudinal or genetically informed designs (e.g., twin studies). Prospective work linking trait-like sensory cortical hyperactivity to later clinical outcomes such as PTSD or autism would be especially valuable for identifying sensory-driven pathological mechanisms. Additionally, recent advances in parameterizing EEG[70] have spurred a surge of research leveraging the aperiodic 1/f slope as a proxy for cortical excitation–inhibition balance across diverse domains, including cognition, arousal, development, aging, and psychiatric disorders[71-76]. However, the 1/f slope remains an indirect index; combining it with complementary measures such as glutamate/GABA ratios derived from magnetic resonance spectroscopy (MRS) could strengthen inference, though MRS captures total rather than synaptically released pools and may not directly reflect functional activity. Finally, as our study focused on cortical VEPs, upstream subcortical contributions remain unexplored. Emerging evidence of emotion-related processing in retinal ganglion cells[77-79] and the visual thalamus[80,81] suggests that pre-cortical mechanisms may also shape early cortical biases in trait anxiety.

In summary, we provide empirical evidence for a direct connection between anxiety and sensory traits that arises at the beginning of cortical processing in primary and secondary sensory cortex. That the brain instantiates adaptable sensory regulation at an upstream, low-order post for external inputs has considerable relevance in relation to mental health. It opens the possibility that a fundamental neural mechanism underlying psychopathology involves early sensory pathophysiology[6,10,13,29], promoting a trifecta conceptualization of psychiatric conditions that spans sensory, emotional, and cognitive domains. Further research is essential to elucidate genetic and environmental factors involved in the development of this intertwined sensory-emotion trait.

## Methods

### Experiment 1

**Participants.** As part of a larger study previously reported in ref. 82, we conducted an EEG experiment of visual discrimination and aversive conditioning consisting of three recording sessions separated by 15 (± 3.6) days. A total of fifty-two individuals (19.5 ± 1.4 years; 22 men) participated in the first two sessions that took place on Day 1, and forty-two returned to participate in the third session on Day 16. Five and six participants were excluded from analysis for Day 1 and Day 16, respectively, due to excessive eye movements, severe EEG artifacts and technical problems, resulting in final samples of 47 and 36 for Day 1 and Day 16 VEP analyses, respectively. All participants were right-handed with normal or corrected-to-normal vision and denied a history of severe head injury, psychological/neurological disorders or current use of psychotropic medication. All participants provided informed consent to participate in this study, which was approved by the University of Wisconsin Institutional Review Board. Participants received course credits or monetary compensation. Sample size was determined based on a medium effect size typically associated with trait anxiety, e.g., its correlations with behavioral and neural measures, as reported in both meta-analyses[83] and our previous research[54,84,85]. Specifically, assuming a medium effect size ($r = 0.35$), a sample size of 46 is required to achieve 80% power.

**Trait anxiety assessment.** At the beginning of the first session, participants completed the BIS, a 7-item self-report measure (rated on a Likert scale of 1–4), assessing trait anxiety or the strength of the behavioral inhibition system[40,54,84,85]. This scale is neurobiologically

motivated with high reliability and strong predictive validity of anxiety[41,42] and recommended by the National Institute of Mental Health for assessing trait anxiety. Importantly, the trait of behavioral inhibition is found to be especially related to sensory sensitivity[43].

**Stimuli.** Two types of Gabor patches (sinusoidal gratings multiplied by a Gaussian envelope; 9° × 9° in visual angles) were generated with specific visual properties that preferentially stimulate the magnocellular (M) and parvocellular (P) pathways, respectively. M-selective Gabor patches were of low spatial frequency (LSF; 0.67 cycles per degree/cpd), low luminance contrast (6.9% Michelson contrast), and achromatic. The lightest point of the Gabor patch was 23.14 cd/m² and the darkest point was 20.16 cd/m². P-selective Gabor patches were of high spatial frequency (HSF; 4 cpd) and chromatic (red-green isoluminant). By making the P-selective stimuli isoluminant, we would maximally inhibit the magnocellular activation evoked by the onset of these stimuli.

Individual red-green isoluminant points were experimentally determined by heterochromatic flicker photometry[86], with gray, red, or green squares alternatingly presented on a CRT monitor at a frequency of 30 Hz. With the luminance of the gray square fixed at 21.65 cd/m² (the same as the background luminance of the experimental tasks), participants first adjusted the intensity of the red gun, via button pressing, until minimal flicker was perceived between the alternating gray and the red squares. With the luminance of the red square set, participants then adjusted the intensity of the green gun until minimal flicker was observed between the alternating red and green squares. After three repetitions, mean red and green values were computed for each participant to generate individualized red-green isoluminant chromatic Gabor patches.

Throughout the experiment, visual stimuli were presented on the same gray background on a CRT monitor, which was calibrated by first fitting a gamma function for each RGB channel based on sampled luminance values of each channel measured by a photometer, and then applying a reverse-gamma function on each RGB channel to achieve uniform steps of luminance increase using the Psychophysics Toolbox[87]. Stimulus presentation was linked to the refresh rate (60 Hz) of the CRT monitor and delivered using Cogent2000 software (Wellcome Dept., London, UK) as implemented in Matlab (Mathworks, Natick, MA). Synchronization between stimulus display and data acquisition was verified using a photodiode placed at the center of the monitor screen.

**Experiment procedure.** The experiment consisted of four main phases: pre-conditioning (*Time 1*), conditioning, Day 1 post-conditioning (*Time 2*), and Day 16 post-conditioning (*Time 3*). Participants were seated ~60 cm from a CRT monitor in a dimly lit, electrically shielded room, and performed heterochromatic flicker photometry to determine their individual red-green isoluminant values, which were used to generate P-selective Gabor patches.

During conditioning, M- and P-selective CS were presented in two separate blocks, with block order counterbalanced. Within each conditioning block, 20 trials (10 for CS+, 10 for CS−) were presented. For 70% of the trials that were reinforced, a CS Gabor patch was centrally presented for 3000 ms, followed by simultaneous delivery of a UCS image for 2000 ms (fearful for CS+, neutral for CS−) and UCS sound for 1500 ms (fearful scream for CS+, pure tone for CS−). For the remaining 30% of trials, no UCS was presented following the CS. Before, immediately after, and 15 days after conditioning, participants performed an orientation discrimination task (see below).

For both M- and P-selective Gabor patches, two orientations were selected and differentially conditioned as CS+ and CS- via pairing with either aversive unconditioned stimuli (UCS) or neutral stimuli, respectively. To avoid orientation-specific effects, we employed two sets of Gabor patches (33° and 57° or 123° and 147° clockwise from the vertical meridian), which were counterbalanced across participants. The assignment of CS+ orientation within each set was further counterbalanced. Aversive and neutral UCS consisted of the simultaneous presentation of images and sounds. Fearful and neutral images were selected from the International Affective Picture Set (IAPS[88]) and internet sources, depicting threatening scenes (e.g., knife put to throat; gun pointed to head) and household artifacts (e.g., whistle, cabinet), respectively. Fearful sounds (i.e., screams) were obtained from the fear subset of human affective vocalizations[89]. Neutral sounds consisted of pure tones of various frequencies.

**Orientation discrimination task (ODT).** The orientation discrimination task (ODT) required participants to detect orientation differences between two serially presented Gabor patches. Each trial began with a centrally presented fixation cross with a jittered duration of 1600-1900 ms. The 1st Gabor patch appeared for 400 ms, followed by a visual mask of 100 ms, before the 2nd Gabor patch appeared for 400 ms, followed by another visual mask of 100 ms. The 1st Gabor patch was always a CS (CS+ or CS-), while the 2nd Gabor patch was either the same or different in orientation compared to the 1st Gabor patch. "Same" or "Different" trials were presented with equal probability. On "Different" trials, a Gabor patch 12° apart from both CS+ and CS- orientations (45° or 135°), was presented. Participants were required to make a speeded judgment to indicate whether the two serially presented Gabor patches were of same or different orientations. A total of 360 trials were randomly presented across three mini-blocks, with 90 trials for each experimental condition (i.e., M/P-selective CS + /CS-). In the current analysis, we pooled the CS+ and CS- trials, yielding 180 trials each for M- and P-selective Gabor patches.

**EEG recording and analysis.** EEG data were continuously recorded during all four phases of the experiment from a 96-channel (BioSemi ActiveTwo) system at a 1024 Hz sampling rate. Electrooculogram (EOG) was recorded at two eye electrodes at the outer canthi of each eye and one infraorbital to the left eye. EEG data were down-sampled to 256 Hz, digital bandpass filtered from 0.1 to 40 Hz, and then referenced to the average of all 96 channels. EEG artifact detection and removal were achieved by the *Fully Automated Statistical Thresholding for EEG artifact Rejection* (FASTER) algorithm implemented in EEGLAB 2021.1[90]. FASTER first interpolates deviant channels from the continuous data using the EEGLAB spherical spline interpolation function. Data were then segmented into epochs 200 ms prior to and 300 ms following the onset of the 1st Gabor patch during orientation discrimination trials of each ODT. Epochs were rejected if their z-scores exceeded ±3 within parameters of amplitude range, variance, and deviation. Epoched data were then re-referenced to the average reference and submitted to independent component analysis (ICA) decomposition using the Infomax algorithm[91]. Artefactual components (i.e., muscular artifacts, eye blinks and saccades, electrode "pop-offs", etc.) were automatically detected and removed from the data. Lastly, deviant channels within individual cleaned epochs were interpolated again using the EEGLAB spherical spline interpolation function. Baseline correction was applied to the 200 ms pre-stimulus period.

We focused on visual event-related potential (VEP), the P1 and C1-N1 complex, which reflect early visual cortical processing and serve to dissociate magnocellular and parvocellular processing[35,36]. At site Oz, for all three time points, we extracted mean P1 amplitudes to M-selective and C1/N1 amplitudes to P-selective stimuli during 113–148 ms and 74–152 ms, respectively, centered on their peak latencies.

## Experiments 2 and 3
**Participants.** Fifty-two right-handed individuals (20.0 ± 3.18 years of age; 16 men) with normal or corrected-to-normal vision participated in

this study, which consisted of a resting-state phase and *Experiments 2 and 3*. All participants denied a history of severe head injury, psychological or neurological disorders, or current use of psychotropic medication. All participants provided informed consent to participate in this study, which was approved by the Florida State University Institutional Review Board. Participants received course credits or monetary compensation. As in *Experiment 1*, based on the medium size effect of trait anxiety, we included 52 participants to ensure sufficient statistical power after potential exclusion due to artifacts and attrition. All 52 participants underwent *Experiments 2 and 3* and were included in the VEP analysis. Of the 44 participants who completed resting-state recordings, one was excluded due to a technical problem, yielding a final sample of 43 for the resting-state analysis.

**Trait anxiety assessment.** As in *Experiment 1*, we administered the BIS questionnaire to assess trait anxiety.

**Stimuli.** Two types of Gabor patches (sinusoidal gratings multiplied by a Gaussian envelope) were included, which contained visual properties that selectively stimulate the magnocellular (M) and parvocellular (P) pathways. In *Experiment 2*, M-selective Gabor patches were of LSF (0.67 cpd), low luminance contrast (6.9% Michelson contrast), and achromatic, which were presented 10° horizontally off-center (extending a 4° × 4° visual angle). P-selective Gabor patches were of HSF (3 cpd), chromatic (red-green) and presented at the center (extending a 4° × 4° visual angle). These patches were made isoluminant by fixing the red and green luminance and the gray background luminance at 15.4 cd/$m^2$, such that magnocellular activation induced by the onset of these stimuli would be maximally inhibited.

In *Experiment 3*, stimuli consisted of elongated ovals (extending a visual angle of 4.8° × 9°) foveally presented in four different orientations (30°, 60°, 150°, 120° clockwise). Two types of achromatic filling gratings (tilting at 30°, 60°, 120°, 150° clockwise) were included: M-selective gratings were of LSF (.67 cpd) and low contrast (6.9% Michelson contrast) and P-selective gratings of HSF (3 cpd) and high contrast (34% Michelson contrast). As in *Experiment 2*, these physical properties were chosen to selectively activate the M- and P-selective pathways.

Stimulus presentation was linked to the refresh rate (60 Hz) of a CRT monitor and delivered using Cogent 2000 software (Wellcome Department, London, UK), as implemented in MATLAB (MathWorks, Natick, MA). Synchronization between stimulus display and data acquisition was verified using a photodiode placed at the center of the monitor screen.

**Experiment procedure.** Participants were seated ~60 cm from a CRT monitor in a dimly lit and electrically shielded room. Eyes-open resting-state EEG was first recorded when participants were asked to fixate on a crosshair on the screen. Participants then performed *Experiments 2 and 3* the sequence of which was counter-balanced across participants.

In *Experiment 2*, a trial began with a blank screen for 100 ms, followed by a Gabor patch for 300 ms and then a gray fixation crosshair with a jittered duration of 700–1000 ms. No response was required from the participants, and twelve catch trials were included to ensure attention to the stimuli. The Gabor patches were either P-selective and presented centrally (100 trials) or M-selective and presented in the left/right visual periphery (10° off-center; 100 trials for each side). During a catch trial, a yellow fixation crosshair was presented for 1300 ms, to which participants were required to make a button press.

In *Experiment 3*, participants performed a similar orientation detection task as in *Experiment 1*. Each trial began with a centrally presented fixation crosshair with a jittered duration of 1600–1900 ms. The 1st oval appeared for 400 ms, followed by a visual mask of 100 ms,

and the 2nd oval appeared for 400 ms, followed by another visual mask of 100 ms. The 2nd oval could be in the same orientation as the 1st oval or differ by 7° or 15°, and participants were required to make a speeded "Same" or "Different" response. A total of 560 trials, 280 trials each for M- or P-selective gratings, were randomly intermixed and presented across four blocks.

**EEG recording and analysis.** Recording procedures and analysis for *Experiments 2 and 3* were largely the same as in *Experiment 1*. In *Experiment 2*, P1 potentials were evoked by M-selective Gabor patches presented in the right or left visual periphery, maximally distributed at the contralateral posterior temporal sites (P7 and P8). We extracted mean P1 amplitudes (collapsed across the two sites) during 125–160 ms at P7 and 121–156 ms at P8 (centered on the respective peak latencies). A C1-N1 complex was evoked by P-selective Gabor patches, maximal at Oz. We extracted mean C1 amplitudes during 62–86 ms interval at Oz (centered on the peak latency). In *Experiment 3*, P1 potentials were evoked by M-selective gratings and C1 potentials by P-selective gratings, both maximal at Oz. We extracted mean P1 and C1 amplitudes during 86–121 ms and 54–70 ms, respectively, centered on their peak latencies.

Resting-state EEG/EOG data were recorded for 2 min using the same recording protocol as described in *Experiment 1*. As in our previous studies[29,92], resting-state EEG data were downsampled to 256 Hz, high-pass (1 Hz) and notch (60 Hz) filtered, and re-referenced to the average of all EEG channels before submission to the FASTER algorithm for further artifact correction. Next, resting-state data were segmented into 1-s epochs and converted into a power spectrum with a Multitaper power spectral density estimate in the signal processing toolbox in Matlab 2021b (The MathWorks, Inc., Natick, Massachusetts, United States). Zero padding was applied to provide a frequency resolution of 0.25 Hz in the 1 s epochs in time-domain EEG data. The aperiodic parameters (e.g., 1/f spectral exponents) of the resting-state power spectrum (3–50 Hz) were calculated using the "Fooof" toolbox 1.0.0 (https://github.com/fooof-tools/fooof[70]). The average $R^2$ of spectral fits was 0.97, with all individual $R^2$ exceeding 0.91, indicating strong overall and individual-level fit quality. According to[70], we extracted aperiodic 1/f spectral exponents from the Cz site (collapsed across the Cz and four surrounding electrodes).

**Exact low-resolution electromagnetic tomography (eLORETA).** Based on VEP amplitudes from hdEEG in *Experiment 2, 3* we performed intracranial source analyses through exact low-resolution electromagnetic tomography (eLORETA v20240713;[44]). The eLORETA algorithm on hdEEG data has been increasingly used for intracranial source estimation in our lab and others[82,84,93–97], having been cross-validated in multiple studies combining EEG-based LORETA with fMRI[98–102], positron emission tomography[103,104], and intracranial recordings[105].

Our solution space for intracranial source analyses consisted of 6239 cortical gray matter voxels with a spatial resolution of 5 × 5 x 5 mm in a realistic head model[106], which was registered to standardized space using a digitized MRI from the Montreal Neurological Institute (MNI). We estimated voxelwise current density during the C1 and P1 windows, which was then submitted to voxelwise correlation with the aperiodic exponent (1/f slope). To minimize false-positive results in intracranial source localization, our laboratory has routinely applied two constraints in the analyses[52,82,84,93,97]. First, eLORETA analysis was only applied to the time windows and tests that were significant in surface-level analysis[107]. Accordingly, we only submitted the C1 sources to the analysis. Second, we applied a Monte Carlo simulation based on the voxel spatial correlation inherent in the data to determine the statistical threshold of corrected $p < 0.05$ FDR. Specifically, using the Gaussian filter widths estimated from our data (FWHMx = 2.57 mm, FWHMy = 2.56 mm, FWHMz = 2.38 mm), the voxel size (5 × 5 x 5 $mm^3$), and a connection radius (5 mm), we derived a

corrected threshold consisting of a voxel-level $p < 0.005$ over five contiguous voxels. All coordinates are reported in MNI space.

## Experiment 4

**Participants.** This dataset was initially reported in ref. 52. Forty-six right-handed individuals ($19.3 \pm 2.3$ years of age; 22 men) with normal or corrected-to-normal vision participated in the study. Six participants were excluded due to excessive eye movements and technical problems, yielding a final sample of 40 participants for VEP analysis. All participants denied a history of severe head injury, psychological or neurological disorders, or current use of psychotropic medication. All participants provided informed consent to participate in this study, which was approved by the University of Wisconsin Institutional Review Board. Participants received course credits or monetary compensation.

**Trait anxiety assessment.** As in *Experiments 1–3*, we administered the BIS questionnaire to assess trait anxiety.

**Stimuli.** Twenty-seven images were selected from the International Affective Picture Set (IAPS[88]) and internet sources, depicting natural scenes/objects of fearful, disgusting, or neutral content (nine for each emotion category). These images were transformed to grayscale and equal size ($256 \times 256$ pixels) and were low-pass filtered at 3 cycles/degree or high-pass filtered at 7 cycles/degree to generate low spatial frequency (LSF; M-selective) or high spatial frequency (HSF; P-selective) images, respectively. Spatial-frequency-filtered images were further normalized to equal luminance ($17.11 \, cd/m^2$) and luminance contrast by the SHINE Matlab toolbox[108]. Next, wavelet analyses[109] were applied to filtered image sets to ensure equal wavelet energy for each emotion category in both H/LSF bands[52]. Similarly, objective measures of visual complexity (edge density, entropy, compressed image size) were extracted[110] and submitted to ANOVAs (Emotion by SF), which revealed no effect of emotion or emotion-by-SF interaction[52].

Stimulus presentation was linked to the refresh rate (60 Hz) of a CRT monitor and delivered using Cogent 2000 software (Wellcome Department, London, UK) as implemented in MATLAB (MathWorks, Natick, MA). Synchronization between stimulus display and data acquisition was verified using a photodiode placed at the center of the monitor screen.

**Procedure.** Participants were seated ~120 cm from a CRT monitor in an electrically shielded room and performed a visual search task adapted from a previous study[84]. Each trial began with a centrally presented fixation crosshair with a jittered duration of 1150–1650 ms, followed by an image ($7.2° \times 7.2°$) centrally displayed for 150 ms. Next, a search array in green was superimposed on the image for 500 ms, consisting of one horizontal bar (target) and seven vertical bars (distracters). Participants were required to make a button press to indicate the quadrant where the target was located while maintaining fixation. There was a total of 600 randomly intermixed trials presented in four experimental blocks, with 100 per each experimental condition (M/P-selective X Fear/Disgust/Neutral).

**EEG recording and analysis.** EEG were recorded from a 96-channel (BioSemi ActiveTwo) system at a 1024 Hz sampling rate, down-sampled to 256 Hz, and digitally bandpass filtered from 0.1 to 40 Hz. Electrooculogram (EOG) was recorded at two eye electrodes at the outer canthi of each eye and one infraorbital to the left eye. The epoch was segmented from −200 to 300 ms post stimulus onset to focus on early visual processing. Trials with EEG/EOG voltages exceeding $\pm 75 \, \mu V$ (relative to pre-stimulus baseline) were excluded from analysis. Inspection of grand average waveforms indicated clear P1s elicited by both M- and P-selective scenes, maximal at the central occipital site Oz,

peaking at 119 ms and 115 ms post-stimulus, respectively. At site Oz, we extracted mean P1 amplitudes in the window of 100–135 ms, centered on the averaged peak latency of the two types of images.

## Reporting summary

Further information on research design is available in the Nature Portfolio Reporting Summary linked to this article.

## Data availability

Source data are provided with this paper (https://github.com/LiLabFSU/Anxious-Vision-Trait-like-cortical-hyperactivity-in-trait-anxiety). The data generated in this study have been deposited in the Github database that is publicly accessible. Source data are provided with this paper.

## Code availability

Code is provided with this paper (https://github.com/LiLabFSU/Anxious-Vision-Trait-like-cortical-hyperactivity-in-trait-anxiety). Code is publicly accessible. All display items presented in the main manuscript and supplementary information can be reproduced from the data and code shared in the public repository.

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

## Acknowledgements

This research was supported by the National Institutes of Health grants (R01MH132209 and R01NS129059, W.L.).

## Author contributions

Z.W. contributed to data analysis and manuscript preparation; Y.Y. contributed to experiment design, data collection, analysis, and manuscript preparation; J.A.B. contributed to data analysis and manuscript preparation; R.J.D. contributed to manuscript preparation; W.L. contributed to experiment design, data analysis, and manuscript preparation.

## Competing interests

The authors declare no competing interests.
