## [Transparent Peer Review file · Nature Communications]

Trait-like visual cortical hyperactivity in trait anxiety

Corresponding Author: Professor Wen Li

Version 0:

Reviewer comments:

Reviewer #1

(Remarks to the Author)

The study tested the hypothesis of early visual cortical responses (ERPs) as a neural marker of trait anxiety. Overall, the study is well motivated, the methods are sound, and, with the caveat that multiple comparisons were not considered in statistics, the findings support the conclusions. Authors should be commended for careful consideration about assessment of individual anxiety and effort in distinguishing the M and P pathways of visual cortical responses. This reviewer has a few comments authors may wish to address in a revision.

Introduction:

1) "A widespread assumption is that sensory cortex maintains high fidelity with respect to external stimuli. This implies its responses should be relatively consistent across individuals—unless modulated by higher-order (limbic and prefrontal) inputs. However, this conflicts with evolutionary evidence that sensory cortex emerged before limbic and prefrontal structures, and that ancient organisms (such as the ancestral amniote) depended on it to flexibly identify predators and prey."

It's not clear how sensory fidelity conflicts with said evolutionary evidence. The statements need to be elaborated on to be convincing.

Methods and Results:

1) "Sample size was determined based on a medium effect size typically observed for trait anxiety (i.e., $r = .35$, requiring a sample size of 46 to reach a power of .80)."

It's not clear how this came about - the effect size of? To truly estimate the power, more details of these earlier studies would need to be provided for it to be convincing.

2) "... while consistency with Time 3 would attest to long-term reliability."

I wouldn't say that assessment across a few weeks is relevant to long-term reliability. It's more about test retest stability.

3) Fifty-two (52) subjects participated in the study. However, from the scatter/regression plots, the n seemed short - were some subjects excluded in data analyses? If so, exclusion criteria need to be specified and the sample size for each set of analyses clearly specified.

4) women and men are known to have different clinical manifestations and potentially pathophysiological processes of anxiety, clinical or subclinical. It would significantly enhance the impact of the study if the analyses are also conducted separately for women and men, with sex-specific findings cross-validated, if found.

5) Somewhat concerning is that the majority of the regressions were significant at $p < 0.05$, without correction for multiple comparisons.

6) Authors estimated the source of ERPs, but this did not appear to be performed for all experiments?

7) Source estimation appeared to show some ERPs in the cuneus and others in the lingual gyrus. Lingual gyrus has been

broadly implicated in emotion processing and in the mechanisms of emotional disorders. Authors may want to carefully consider this issue to give justice to the data.

Reviewer #2

(Remarks to the Author)

The study investigates the relationship between trait anxiety and early visual cortical hyperactivity, focusing on the parvocellular (P) pathway in the visual system. The authors report that individuals with high trait anxiety exhibit heightened activity in the primary visual cortex (V1/V2) as early as 50-100 ms post-stimulus, specifically in the P-pathway (but not the magnocellular/M-pathway). This hyperactivity is stable across time, arousal states, and stimulus types (ranging from basic Gabor patches to complex natural scenes). Furthermore, in low-anxiety individuals, the aperiodic exponent (1/f slope) of resting-state EEG (a proxy for E/I ratio) predicts P-pathway response magnitude, suggesting intact E/I modulation. In high-anxiety individuals, this relationship breaks down, implying impaired E/I regulation as a potential mechanism for hyperactivity. Overall, I find this study well-conceived and clearly written. However, closer inspection of the data related to E/I (de)regulation as a mechanism suggests the preliminary nature of this observation. The manuscript might be better suited for a more specialized journal where additional methodological details could be explored.

The paper has several strengths. First, it features a robust and reproducible design, with a relatively large sample size across four experiments. The findings also control for state anxiety through aversive conditioning and address stimulus confounds (e.g., luminance, spatial frequency). Second, the results shift focus away from frontal-limbic dysfunction in anxiety, highlighting instead early sensory changes associated with anxiety. Third, the experimental design and stimulus presentation appear rigorous, setting this work apart from many similar studies in the field.

However, there are also clear weaknesses. First, the evidence supporting E/I imbalance as a mechanism remains limited. For instance, the key E/I finding relies on a one-tailed post-hoc test ($p = 0.04$), which is statistically weak and risks Type I error. Moreover, the aperiodic exponent (1/f slope) serves as only an indirect measure of E/I balance; concurrent fMRI/MRS or pharmacological manipulations would strengthen these claims. Without additional corroborating evidence, the current conclusions should be considered preliminary.

Second, scalp EEG has poor spatial resolution, making it difficult to definitively localize activity to V1/V2. While eLORETA provides some assistance, it remains an estimation - I find it particularly challenging to achieve such precise localization, even with MEG recordings using >160 channels.

Third, the current evidence remains purely correlational. There is no causal evidence (e.g., TMS or lesion studies) to confirm whether V1 hyperactivity is necessary for anxiety. Additional TMS evidence targeting V1/V2 stimulation could substantially strengthen the argument for early sensory deficits and regional specificity.

Fourth, the M-pathway hypoactivity observed in Experiment 1 was not clear in Experiments 2-4, raising questions about its robustness. The reciprocal P/M relationship (Fig. S1) might reflect task design (e.g., rapid alternation suppressing M responses) rather than being a genuine trait anxiety effect.

Finally, while trait anxiety represents a risk factor for disorders (e.g., PTSD, autism), the study doesn't examine clinical populations or establish causal links to symptoms. It remains unclear whether anxiety leads to hypersensitivity in sensory regions or whether sensory hypersensitivity contributes to anxiety development.

Reviewer #3

(Remarks to the Author)

The paper by Wu and colleagues describes a set of experiments, two new and two previously published from a different perspective/scope, that reveal trait anxiety-induced hyperactivity in the visual cortex. This hyperactivity manifests very early, starting at 46 ms in one of the experiments. The authors interpret these results as showing that anxiety involves mechanisms that are not mediated in a top-down fashion, but occur in the "earliest processing phase", before those top-down processes take place. The idea that structures higher in the processing hierarchy (amygdala, hippocampus, limbic system or prefrontal regions, as mentioned by authors) are not necessarily involved in all neural mechanisms underlying trait anxiety is very interesting and I fully agree with it. However, this article presents several major problems, in my opinion.

1-On the exclusive allocation of effects to the visual cortex and its status as the "earliest processing stage".

One of the key conclusions of the paper is that visual cortex activity is affected by trait anxiety, and that this cortical region is solely responsible for the observed effects. In relation to this, and throughout the paper, the idea that visual processing begins in the visual cortex is consistently defended. For example, at several (relevant) passages of the paper, such as the abstract or the conclusion paragraph, the authors indicate that the visual cortex is the "earliest processing stage" or "the earliest staging post for external inputs".

However, every pre-cortical key structure within the visual ascending route, including the retinal ganglion cells and the visual thalamus, are active visual processors. For example, retinal ganglion cells change their response to the very same stimulus depending on the top-down modulation they receive (Warwick et al., 2024). Similarly, the response of the visual thalamus (e.g., the lateral geniculate nucleus -LGN-, which is the main upstream modulator of striate visual cortex) to identical stimuli also varies as a function of task demands (e.g., see reviews by Ghodrati et al., 2017; Halassa and Kastner, 2017; Saalman and Kastner, 2011; Weyand, 2016).

Clearly, these structures, which are on the main input way to the visual cortex, strongly influence its activity. In other words,

the effects detected in the visual cortex could be at least partially due to these precortical stages (whose activity cannot be detected directly by ERPs, but indirectly through their effects on the visual cortex). Importantly, this precortical activity may underlie certain types of anxiety (Milosavljevic et al., 2016), or be affected by anxiety or arousal (e.g., Salay, 2021; Schröder et al., 2020).

In the same vein, and crucially, another key finding of the paper is the differential effect of trait anxiety as a function of the two visual streams, parvocellular and magnocellular, which originate in the LGN of the thalamus and reach, without intermediate modulation, the visual cortex at approximately the latency at which the first anxiety effects are reported here.

2- On parvo- and magnocellular activity discrimination based on spatial frequency or luminance/color manipulation.

Please note that manipulation of spatial frequency, and/or luminance and color, does not guarantee the distinction of magnocellular and parvo-cellular visual processing, as assumed in the paper (although this strategy has often been employed). For example, see these works by Bernt Skottun:

Skottun, B. C. (2015). On the use of spatial frequency to isolate contributions from the magnocellular and parvocellular systems and the dorsal and ventral cortical streams. *Neuroscience & Biobehavioral Reviews*, 56, 266-275.

Skottun, B. C. (2013). On using isoluminant stimuli to separate magnocellular and parvocellular responses in psychophysical experiments—A few words of caution. *Behavior Research Methods*, 45(3), 637-645.

3- On the experimental design and logic.

Experiments 2 and 3 were designed *de novo* for this study (Experiments 1 and 4 consist of reanalyzing data from past published experiments, but from a new perspective/scope). The authors indicate that Experiment 3 was designed to rule out that the results of Experiment 2 were due to a restricted sensitivity to color processing, so that the stimuli in Experiment 3 were achromatic. However, the authors also indicate that the order of Experiment 2 and 3 was counterbalanced (the participants were the same in both experiments). In other words, the authors designed Experiment 3 at the same time as Experiment 2. The question that arises is why, if the authors considered from the beginning that color could be an interfering factor, they did not go directly to the experiment without color (current Experiment 3).

4-On the interpretation of results.

Two of the main interpretations/conclusions contradict (at least apparently) some of the main results. On the one hand, the correlations between anxiety scores and P1 amplitude did not differ as a function of the emotional load of the images (neutral, fearful, and aversive): Experiment 4. However, the authors indicate that the described modulation of visual cortex activity by trait anxiety serves to better detect threat (e.g., “hyperactivity in visual cortex could facilitate overactivation of threat memory, intensifying threat detection and recall,” p. 11). If the anxiety-induced increase in visual cortex activity is similar for threatening and neutral stimuli (Exp. 4), this interpretation does not seem justified.

On the other hand, the key result and conclusion pointing to the parvocellular, but not magnocellular, visual processing stream as responsible for the observed effects are also not clearly in agreement with some interpretation in the Discussion section. For example, “In individuals with high trait anxiety, sensory cortex hyperactivity (and disinhibition) would allow irrelevant or unwanted environmental input to evoke sensory cortical responses”. However, irrelevant or unwanted environmental inputs are typically projected onto non-foveal regions of the retina (in other words, gaze does not focus on them). Considering that retinal projections to P and M layers of the LGN decline to a greater extent in the former case with eccentricity (e.g., Brown et al., 2005), resulting in a magnocellular bias for peripheral vision, this interpretation seems also contradictory.

5- Other relevant comments.

- Given that the mean amplitude is very dependent on the width of the analyzed window of ERPs (or window of interest - WOI-), please explain how WOI widths were defined.

- No evidence of effect is not the same as evidence of no effect. The absence of significant correlation between ERP amplitude and anxiety scores is interpreted as a proof in favor of the null hypothesis -i.e., no effect- (e.g., “negating the presence of a magnocellular hypoactivity in trait anxiety”, p. 7). Indeed, and in order to determine whether this conclusion is valid, the likelihood of H_0 should be computed, for example, through Bayesian methods.

References

- Brown, L. E., Halpert, B. A., & Goodale, M. A. (2005). Peripheral vision for perception and action. *Experimental Brain Research*, 165, 97–106.
- Ghodrati, M., Khaligh-Razavi, S., & Lehky, S. R. (2017). Towards building a more complex view of the lateral geniculate nucleus: Recent advances in understanding its role. *Progress in Neurobiology*, 156, 214-255.
- Halassa, M. M., & Kastner, S. (2017). Thalamic functions in distributed cognitive control. *Nature Neuroscience*, 20(12), 1669-1679.
- Milosavljevic, N., Cehajic-Kapetanovic, J., Procyk, C. A., & Lucas, R. J. (2016). Chemogenetic activation of melanopsin retinal ganglion cells induces signatures of arousal and/or anxiety in mice. *Current Biology*, 26(17), 2358-2363.
- Saalmann, Y. B., & Kastner, S. (2011). Cognitive and perceptual functions of the visual thalamus. *Neuron*, 71, 209-223.
- Salay, L. D., & Huberman, A. D. (2021). Divergent outputs of the ventral lateral geniculate nucleus mediate visually evoked defensive behaviors. *Cell Reports*, 37(109792).
- Schröder, S., Steinmetz, N. A., Krumin, M., Pachitariu, M., Rizzi, M., Lagnado, L., ... & Carandini, M. (2020). Arousal modulates retinal output. *Neuron*, 107(3), 487-495.
- Warwick, R. A., Riccitelli, S., Heukamp, A. S., Yaakov, H., Swain, B. P., Ankri, L., ... & Rivlin-Etzion, M. (2024). Top-down modulation of the retinal code via histaminergic neurons of the hypothalamus. *Science Advances*, 10(35), eadk4062.

Reviewer comments:

Reviewer #1

(Remarks to the Author)

Authors have addressed the concerns and suggestions raised in earlier review. I have no further comment. The study represents a significant and innovative addition to the literature of human neuroscience.

Reviewer #2

(Remarks to the Author)

The authors have adequately addressed my concerns. While it remains challenging to infer neuronal-level properties such as excitation/inhibition imbalance from scalp EEG, the authors present a thoughtful research design that advances our understanding of how such underlying processes - and their pathological alterations - might be reflected in scalp EEG signals. I commend the authors for making an important contribution to the field.

Response to the comments on the submission “Anxious vision: Trait-like visual cortical hyperactivity in trait anxiety” by Wu et al (NCOMMS-25-13802)

Responses to Reviewer 1:

We are encouraged that the reviewer commented on our “*novel and interesting approach*” and considered “*the contribution to the field potentially very strong*”. We are also extremely grateful for the excellent comments and constructive suggestions, which have been fully addressed and incorporated in the revised manuscript. Our responses are detailed below. The Reviewer’s original comments are quoted verbatim and displayed in grey shading. Modifications are highlighted in the text and reproduced here in corresponding responses (highlighted in blue).

“The study tested the hypothesis of early visual cortical responses (ERPs) as a neural marker of trait anxiety. Overall, the study is well motivated, the methods are sound, and, with the caveat that multiple comparisons were not considered in statistics, the findings support the conclusions. Authors should be commended for careful consideration about assessment of individual anxiety and effort in distinguishing the M and P pathways of visual cortical responses. This reviewer has a few comments authors may wish to address in a revision.

Introduction:

1) "A widespread assumption is that sensory cortex maintains high fidelity with respect to stimuli. This implies its responses should be relatively consistent across individuals—unless modulated by higher-order (limbic and prefrontal) inputs. However, this conflicts with evolutionary evidence that sensory cortex emerged before limbic and prefrontal structures, and that ancient organisms (such as the ancestral amniote) depended on it to flexibly identify predators and prey." It's not clear how sensory fidelity conflicts with said evolutionary evidence. The statements need to be elaborated on to be convincing."

We thank the Reviewer and have clarified the statement, as reproduced below:

p. 3: A widespread assumption is that the sensory cortex maintains high fidelity to external stimuli, which implies consistent responses across individuals—unless modulated by higher-order prefrontal or limbic inputs. However, the notion of rigid sensory encoding conflicts with the evolutionary history of sensory cortex: Phylogenetic evidence indicates that sensory cortex evolved prior to the emergence of limbic and prefrontal structures, as observed in ancient organisms such as the ancestral amniote (21, 22). Absent these higher-order structures, sensory cortex likely evolved flexible encoding—dynamically shaped by internal states and the biological value of external stimuli—to support adaptive interactions with the environment.

“Methods and Results:

1) "Sample size was determined based on a medium effect size typically observed for trait anxiety (i.e., $r = .35$, requiring a sample size of 46 to reach a power of .80)."

It's not clear how this came about - the effect size of? To truly estimate the power, more details of these earlier studies would need to be provided for it to be convincing.”

We regret the omission of relevant references demonstrating the medium effect size typically associated with trait anxiety, e.g., its correlations with behavioral and neural measures. This information has now been included in the resubmission.

pp.14-15: Sample size was determined based on a medium effect size typically associated with trait anxiety, e.g., its correlations with behavioral and neural measures, as reported both in meta-analyses (84) and our previous research (56, 85, 86). Specifically, assuming a medium effect size ($r = .35$), a sample size of 46 is required to achieve 80% power.

“2) "... while consistency with Time 3 would attest to long-term reliability."

I wouldn't say that assessment across a few weeks is relevant to long-term reliability. It's more about test retest stability.”

The Reviewer raises a fair point, and we agree. As per the Reviewer's suggestion, we have modified this statement:

p. 5: while consistency with Time 3 would attest to **robust test-retest** reliability.

“3) Fifty-two (52) subjects participated in the study. However, from the scatter/regression plots, the n seemed short - were some subjects excluded in data analyses? If so, exclusion criteria need to be specified and the sample size for each set of analyses clearly specified.”

We apologize for the confusion. In the resubmission, we have provided more specific information about exclusion and sample sizes for specific analyses, as reproduced below.

For Experiment 1:

p. 14: A total of fifty-two individuals (19.5 +/- 1.4 years; 22 men) participated in the first two sessions that took place on Day 1, and forty-two returned to participate in the third session on Day 16. Five and six participants were excluded from analysis for Day 1 and Day 16, respectively, due to excessive eye movements, severe EEG artifact and technical problems, **resulting in final samples of 47 and 36 for Day 1 and Day 16 VEP analyses, respectively.**

For Experiments 2 & 3:

p. 18: **All 52 participants underwent Experiments 2 & 3 and were included in the VEP analysis. Of the 44 participants who completed resting-state recordings, one was excluded due to a technical problem, yielding a final sample of 43 for the resting-state analysis.**

For Experiment 4:

p. 21: Six participants were excluded due to excessive eye movements and technical problems, yielding a final sample of 40 participants for VEP analysis.

“4) women and men are known to have different clinical manifestations and potentially pathophysiological processes of anxiety, clinical or subclinical. It would significantly enhance the impact of the study if the analyses are also conducted separately for women and men, with sex-specific findings cross-validated, if found.”

The Reviewer makes an excellent suggestion, which we deeply appreciate. We have now included a new section to report new analyses of potential sex effects (both simple and interactive with anxiety) on the VEPs. Please note that to ensure sufficient power this necessitated, we pooled participants across all three samples for the following three analyses.

1. Using hierarchical multiple linear regression, we entered the VEP as the outcome variable and included Sample (dummy-coded), Sex, and Trait Anxiety (BIS scores) as regressors in the first step and Sex-by-Anxiety in the second step. For the P-selective VEP, we replicated the significant anxiety effect ($p = .005$) but observed no main effect of Sex ($p = .882$) or Sex-by-Anxiety interaction ($p = .728$). For the M-selective VEP, similar to the main results, there was no effect of anxiety or sex.
2. To further probe these effects, we examined the effect of trait anxiety within each sex separately. Anxiety effects on the P-selective VEP were comparable across the sexes, while no anxiety effects emerged for the M-selective VEP in either sex.
3. Finally, we compared trait anxiety levels between the sexes and found that trait anxiety was lower in males (18.68 ± 4.08) than females (20.27 ± 4.23), $t(136) = 2.23$, $p = .027$, confirming the well-established sex bias in anxiety and helping to rule out systematic sampling bias in our study.

In summary, while male and female participants differed in trait anxiety levels, the association between anxiety and P-selective VEPs was consistent across sexes, suggesting this sensory cortical hyperactivity in trait anxiety is independent of sex. The results are reproduced below for the Reviewer's convenience:

pp. 9-10: Sex independence: Similar trait anxiety association with parvocellular visual cortical hyperactivity in women and men

There is a well-established sex difference in prevalence and clinical manifestations of anxiety (55), potentially reflecting distinct pathophysiological processes. On this basis, in hierarchical multiple linear regressions on pooled data from our three independent samples (see SOM for details), we explored whether women and men differ with respect to an association between trait anxiety and visual cortical hyperactivity. Separate models were implemented for P- and M-selective VEPs, with Sample (dummy-coded), Sex, and Trait Anxiety (BIS scores) entered in the first step and Sex-by-Anxiety in the second step as regressors. For the P-selective VEP, we replicated the significant anxiety effect ($sr = .35$, $p < .001$) but observed no Sex-by-Anxiety

interaction ($sr = -.03$, $p = .762$). For the M-selective VEP, there was no effect of anxiety ($sr = -.02$, $p = .864$), consistent with the main results, and no sex-by-anxiety interaction ($sr = -.004$, $p = .960$).

We also performed similar regression analyses in female and male groups separately. In both groups, trait anxiety significantly predicted P-selective VEPs (females/males: $sr = .36/.35$; $p = .001/.011$). By contrast, neither group showed a significant association between trait anxiety and M-selective VEPs (females/males: $sr = -.03/-.01$; $p = .760/.948$). Notably, a simple t-test on BIS scores showed significantly greater trait anxiety in females (20.27 ± 4.23) compared to males (18.68 ± 4.08), $t(136) = 2.23$, $p = .027$, consistent with the known sex bias in anxiety (55) and suggesting also an absence of systematic sampling bias in our study. In summary, we found no evidence that sex moderates the relationship between trait anxiety and visual cortical hyperactivity, suggesting that this sensory cortical hyperactivity in trait anxiety is independent of sex.

p. 10: Notably, while female participants showed higher levels of trait anxiety than male participants, they exhibited comparable association between trait anxiety and visual cortical hyperactivity, suggesting sex independence of this effect.

“5) Somewhat concerning is that the majority of the regressions were significant at $p < 0.05$, without correction for multiple comparisons.”

We regret that exact p values were not reported in the initial submission. In the resubmission, exact p values are now provided. Importantly, concerning the key finding—an association between trait anxiety and P-selective hyperactivity—six correlational analyses were conducted (3 in **Experiment 1** and 1 each in **Experiments 2-4**). As shown in the histogram below, the p values were largely around 0.01: $p = 0.011$, 0.009, & .026 (**Experiment 1**); $p = 0.018$ (**Experiment 2**); $p = 0.006$

(**Experiment 3**); and $p = 0.017$ (**Experiment 4**). In addition, while the initial manuscript did not perform correction for the multiple tests—given that the tests were *a priori* hypothesis-driven—we agree with the Reviewer that rigor of our study can be further improved by applying such correction. On this basis, we submitted these p values to Benjamin-Hochberg correction, which confirmed that all remain significant at the corrected threshold (q^*):

i	p	q*	p < q*
1	0.0060	0.0083	TRUE
2	0.0090	0.0167	TRUE
3	0.0110	0.0250	TRUE
4	0.0170	0.0333	TRUE
5	0.0180	0.0417	TRUE
6	0.0260	0.0500	TRUE

p. 9: To ensure rigor and robustness of the findings, we further submitted the six *p*-values from the six correlational tests between trait anxiety and P-selective VEPs, across **Experiments 1-4**, to Benjamin-Hochberg correction for multiple comparisons. All six remained significant at the adjusted thresholds (see SOM).

Additionally, please note we conducted data-driven, point-by-point analyses across the entire ERP window (0–300 ms), with correction for multiple comparisons. Confirming the main results above, these analyses identified significant intervals that mapped closely onto the VEP windows (see grey boxes in **Fig. 1B-D**; **Fig. 2B & D**; **Fig. 3B**; and SOM).

“6) Authors estimated the source of ERPs, but this did not appear to be performed for all experiments?”

The original submission focused on demonstrating the site of E/I modulation of VEPs and localized it to the early visual cortex (V1/V2).

The resubmitted SOM now includes source-level results for the association between trait anxiety and VEPs in all experiments. Given the limited spatial precision of EEG source localization, we note that these analyses are largely confirmatory—serving to confirm the well-established source of C1 in early visual cortex, particularly V1 (Clark and Hillyard 1996; Martinez et al. 1999) and likely also V2 (Foxe and Simpson 2002; Ossenblok and Spekreijse 1991). As shown in **Figs. S2-S4** (reproduced below), despite some variability in the exact loci—partly due to the limited spatial precision of EEG—the association between anxiety and P-selective VEPs was consistently localized to early visual cortex, primarily V1/V2.

p. 5: Finally, leveraging the high-density EEG recordings, we also performed intracranial source-level analyses on the C1 and P1 components using exact low-resolution electromagnetic tomography (eLORETA) (46), which confirmed the involvement of early visual cortex (V1/V2; see SOM and **Fig. S2**).

pp. 6-7: In addition, source-level analysis in eLORETA further confirmed an association within early visual cortex (V1/V2; see SOM and **Fig. S3 Left**).

p. 7: In addition, source-level analysis in eLORETA further confirmed an association within early visual cortex (V2; see SOM and Fig. S3 right).

p. 9: In addition, source-level analysis in eLORETA further confirmed an association within early visual cortex (V1/V2; see SOM and Fig. S4).

SOM p. 8: It is well-established that the C1 component originates from the earliest regions of the visual cortical hierarchy, particularly V1 (14, 15) and likely also V2 (16, 17). In keeping with that, our source-level analyses across all four experiments consistently revealed significant associations between trait anxiety and P-selective VEPs in early visual cortex (V1/V2).

Figure S2 Early visual cortex localized for association between trait anxiety and VEPs (Experiment 1; display threshold $p < .005$ uncorrected)

Figure S3 Early visual cortex localized for association between trait anxiety and P-selective C1 (Experiments 2 & 3; display threshold $p < .005$ uncorrected)

Figure S4 Early visual cortex localized for associations between trait anxiety and P-selective P1 (Experiment 4; display threshold $p < .005$ uncorrected)

“7) Source estimation appeared to show some ERPs in the cuneus and others in the lingual gyrus. Lingual gyrus has been broadly implicated in emotion processing and in the mechanisms of emotional disorders. Authors may want to carefully consider this issue to give justice to the data.”

The Reviewer makes a keen and accurate observation—indeed, the lingual gyrus (BA 18/V2), was identified in a few source-level analyses here. It is well-established that the C1 component originates from the earliest regions of the visual cortical hierarchy, particularly V1 (Clark and Hillyard 1996; Martinez et al. 1999) and likely also V2 (Foxe and Simpson 2002; Ossenblok and Spekreijse 1991). Accordingly, our source analysis aimed to confirm the involvement of V1 and/or V2. We greatly appreciate the Reviewer’s acknowledgment of prior research (including our work) implicating the lingual gyrus in emotion processing, particularly among individuals with heightened anxiety (Krusemark & Li, 2011, 2013; Li et al., 2008). Moreover, recent findings including our own have also identified threat processing in the V1 (Li & Keil, 2023; Li et al., 2019; You et al., 2021). Representing a significant advance beyond prior findings, the current study reveals a trait-like, general (non-emotion-specific) hyperfunctioning of early visual cortex (including both V1 and V2) associated with trait anxiety. We have now incorporated this discussion in the resubmission.

SOM p. 8: It is well-established that the C1 component originates from the earliest regions of the visual cortical hierarchy, particularly V1 (14, 15) and likely also V2 (16, 17). In keeping with that, our source-level analyses across all four experiments consistently revealed significant associations between trait anxiety and P-selective VEPs in early visual cortex (V1/V2). While it is known that secondary visual cortex, such as the lingual gyrus in V2, can support the processing of emotional stimuli (9, 12, 18), recent findings, including our own, have also implicated V1 in threat processing (10, 19, 20). Representing a significant advance beyond prior findings, the current study indicates that early visual cortex (V1 and V2) not only flexibly encodes emotional value in sensory input but also exhibits a dispositional, general hyperfunctioning associated with trait anxiety.

Responses to Reviewer 2:

We are encouraged the Reviewer considers our design “*robust and reproducible*” and “*rigorous*” and affirms our focus on “*early sensory changes*” (“*away from the frontal-limbic dysfunctions*”). We also greatly appreciate the Reviewer’s thoughtful comments and useful suggestions, which have been fully addressed—through new analyses and extensive revisions—and incorporated in the revised manuscript. We believe the manuscript is substantially improved following receipt of these excellent, constructive inputs. Our responses are detailed below. The Reviewer’s original comments are quoted verbatim and displayed in grey shading. Modifications are highlighted in the text and reproduced here in corresponding responses (highlighted in blue).

“The study investigates the relationship between trait anxiety and early visual cortical hyperactivity, focusing on the parvocellular (P) pathway in the visual system. The authors report that individuals with high trait anxiety exhibit heightened activity in the primary visual cortex (V1/V2) as early as 50-100 ms post-stimulus, specifically in the P-pathway (but not the magnocellular/M-pathway). This hyperactivity is stable across time, arousal states, and stimulus types (ranging from basic Gabor patches to complex natural scenes). Furthermore, in low-anxiety individuals, the aperiodic exponent (1/f slope) of resting-state EEG (a proxy for E/I ratio) predicts P-pathway response magnitude, suggesting intact E/I modulation. In high-anxiety individuals, this relationship breaks down, implying impaired E/I regulation as a potential mechanism for hyperactivity. Overall, I find this study well-conceived and clearly written. However, closer inspection of the data related to E/I (de)regulation as a mechanism suggests the preliminary nature of this observation. The manuscript might be better suited for a more specialized journal where additional methodological details could be explored.

The paper has several strengths. First, it features a robust and reproducible design, with a relatively large sample size across four experiments. The findings also control for state anxiety through aversive conditioning and address stimulus confounds (e.g., luminance, spatial frequency). Second, the results shift focus away from frontal-limbic dysfunction in anxiety, highlighting instead early sensory changes associated with anxiety. Third, the experimental design and stimulus presentation appear rigorous, setting this work apart from many similar studies in the field.

1). However, there are also clear weaknesses. First, the evidence supporting E/I imbalance as a mechanism remains limited. For instance, the key E/I finding relies on a one-tailed post-hoc test ($p = 0.04$), which is statistically weak and risks Type I error. Moreover, the aperiodic exponent (1/f slope) serves as only an indirect measure of E/I balance; concurrent fMRI/MRS or pharmacological manipulations would strengthen these claims. Without additional corroborating evidence, the current conclusions should be considered preliminary.”

The Reviewer raises several excellent points.

- 1) We observed a significant (two-tailed) interaction between 1/f slope and trait anxiety in predicting C1 ($p = 0.027$). This interaction is directly pertinent to hypothesis testing: trait anxiety interacts with cortical E/I balance to impact early visual cortical processing. To further probe this interaction, we performed secondary analyses—separate correlation analyses between 1/f slope and C1 in the low and high anxiety groups. Given an established directionality in the relation between 1/f slope and neural activity (i.e., steeper slopes reflecting greater inhibition), and the fact subgroup tests were guided and protected by the significant interaction, we considered a one-tailed significance threshold was statistically appropriate.
- (2) Nevertheless, we fully concur with the Reviewer that our results should be interpreted with caution and require replication in larger samples. We have thus toned down our initial claim. We also acknowledge the importance of validating such findings with complementary modalities, such as concurrent fMRI/MRS, to corroborate the VEP and 1/f slope measures (although the sluggish nature of fMRI may limit its sensitivity to early sensory-driven biases). In addition, we wish to note that the field still lacks fully validated markers of E/I balance in humans. MRS-derived measures, such as glutamate (Glu) and GABA concentrations or the Glu/GABA ratio, have recognized limitations and are thus considered *putative* markers. For instance, MRS detects *total* Glu and GABA, not *synaptic* released pools alone; furthermore, these concentrations may not directly reflect neurotransmitter function. Therefore, MRS-based indices of E/I balance can benefit from complementary assessments using other modalities, such as 1/f slope, to boost their validity. We have now included a dedicated limitations section in the revised Discussion, where this point is specifically addressed.

p. 2: suggesting disrupted E/I modulation in trait anxiety **may** underpin this hyperactivity.

pp. 7-8: As discussed above, sensory cortical functioning is regulated by cortical E/I balance, and disruptions in this balance (i.e., E/I imbalance) are implicated in various psychiatric disorders, especially those involving significant sensory anomalies. Thus, we hypothesized that disrupted E/I modulation in high trait anxiety serves as a mechanism underlying its associated sensory cortical hyperactivity. To test this, we recorded resting-state EEG among participants in **Experiments 2 & 3** and extracted an index of E/I balance—the aperiodic exponent (1/f slope)—from the EEG power spectrum (3-50 Hz). **Converging evidence, variously derived from computational modeling, intracranial recordings, and EEG data suggests that the 1/f slope is sensitive to excitatory and inhibitory interplay at the level of the synapse, as well as GABAergic pharmacological modulation (48-52). Importantly, the 1/f slope closely tracks E/I ratios: flatter slopes reflect higher E/I ratios (greater excitation), steeper slopes lower E/I ratios (greater inhibition) (51, 53).**

p. 8: **In support of our hypothesis, we observed a significant interaction between the 1/f slope and trait anxiety in predicting C1 magnitude (coefficient = 0.46, SE = 0.20, $p < 0.05$).**

pp. 13-14: Additionally, recent advances in parameterizing EEG (72) have motivated an upsurge of studies, that avail of the aperiodic 1/f slope as an E/I marker, encompassing domains such as cognition, arousal, development, aging, and psychiatric disorders (73-78). That said, we acknowledge, the 1/f slope remains an indirect index, and its validity should be strengthened by availing of complementary measures. For example, magnetic resonance spectroscopy (MRS) can yield estimates of glutamate (Glu) and GABA concentrations to reflect E/I ratios, albeit MRS captures *total* rather than *synaptic* released pools, and these concentrations may not directly reflect functional activity. Therefore, to bolster the assessment of E/I balance, future research could consider combining MRS-derived Glu/GABA ratios with the EEG 1/f slope.

(3) The Reviewer also raises a great point regarding pharmacological manipulations. We agree that such approaches are effective for demonstrating causal or mechanistic associations. We also concur that causal and mechanistic evidence is important. In the revised Discussion, we now explicitly acknowledge this limitation and emphasize the need for future mechanistic investigation through longitudinal approaches or quasi-experimental designs (e.g., twin studies) to better isolate causality.

We also wish to note that the current study focuses on *traits*—*stable, enduring* dispositions. Even the 1/f slope index was extracted from resting-state EEG to reflect intrinsic, relatively stable E/I balance that may play a role in gradually shaping the association between anxiety and sensory traits. Experimental examination of the causal effects of such traits is inherently challenging, as they are typically stable, endogenous characteristics that are not easily manipulated. Instead, experimental manipulations are likely to introduce transient *state* shifts that could confound trait-level associations. In fact, as the Reviewer kindly noted, we made a special effort to rule out probable state-level confounds (i.e., state anxiety and arousal). Therefore, we tend to think that although pharmacological agents may alter cortical E/I balance, they may also directly cause a state shift in visual cortical activity. The difficulty in disentangling these two effects poses a challenge for isolating and accurately interpreting the association between visual cortical activity and trait anxiety.

p. 13: Several limitations to our studies warrant discussion. Trait research, with its focus on stable, enduring dispositions, inherently limits experimental manipulation and relies primarily on correlational analyses. While interventional techniques, such as pharmacological manipulation and neurostimulation, can alter anxiety or sensory processing, they typically induce transient, state-level shifts that might confound the interpretation of trait-level associations. To establish causality, longitudinal designs or genetically informed approaches (e.g., twin studies) would be particularly useful. Notably, prospective studies that link trait-like sensory cortical hyperactivity to the subsequent development of clinical conditions such as PTSD or autism would be especially valuable for isolating sensory-driven pathological mechanisms. Our hope is that the present study by drawing attention to the underrecognized association between trait anxiety and visual cortical hyperactivity, will stimulate such efforts.

2). “Second, scalp EEG has poor spatial resolution, making it difficult to definitively localize activity to V1/V2. While eLORETA provides some assistance, it remains an estimation - I find it particularly challenging to achieve such precise localization, even with MEG recordings using >160 channels.”

The Reviewer raises another important point. We fully acknowledge the limited spatial resolution of scalp EEG, particularly for endogenous, late ERPs (e.g., P3 or late positive potential) that have distributed sources. However, the current study focuses on exogenous, early visual-evoked potentials—particularly the C1 component—for which, as the Reviewer is no doubt well aware, a substantial body of EEG source-level analyses has consistently identified its *focal* source within early visual cortex (V1/V2), including early classical studies (Clark & Hillyard, 1996; Foxe & Simpson, 2002; Martinez et al., 1999; Ossenblok & Spekreijse, 1991). Importantly, EEG C1 source localization has also been cross-validated using complementary neuroimaging techniques, including fMRI and MEG (Hagler et al., 2009; Martinez et al., 1999). This provides a reassuring background in relation to likely localization.

While the original submission focused on demonstrating the site of E/I modulation of VEPs, to further strengthen the rigor and reproducibility of our source analyses, the resubmission now includes source localization for the association between trait anxiety and VEPs in all four experiments. As shown in **Figs. S2-S4** (reproduced below), despite some variability in the exact loci—partly due to the limited spatial precision of EEG—the association was consistently and robustly localized to early visual cortex, primarily V1/V2.

p. 5: Finally, leveraging the high-density EEG recordings, we also performed intracranial source-level analyses on the C1 and P1 components using exact low-resolution electromagnetic tomography (eLORETA) (46), which confirmed the involvement of early visual cortex (V1/V2; see SOM and **Fig. S2**).

pp. 6-7: In addition, source-level analysis in eLORETA further confirmed an association within early visual cortex (V1/V2; see SOM and **Fig. S3 Left**).

p. 7: In addition, source-level analysis in eLORETA further confirmed an association within early visual cortex (V2; see SOM and **Fig. S3 right**).

p. 9: In addition, source-level analysis in eLORETA further confirmed an association within early visual cortex (V1/V2; see SOM and **Fig. S4**).

SOM p. 8: It is well-established that the C1 component originates from the earliest regions of the visual cortical hierarchy, particularly V1 (14, 15) and likely also V2 (16, 17). In keeping with that, our source-level analyses across all four experiments consistently revealed significant associations between trait anxiety and P-selective VEPs in early visual cortex (V1/V2).

Figure S2 Early visual cortex localized for association between trait anxiety and VEPs (Experiment 1; display threshold $p < .005$ uncorrected)

Figure S3 Early visual cortex localized for association between trait anxiety and P-selective C1 (Experiments 2 & 3; display threshold $p < .005$ uncorrected)

Figure S4 Early visual cortex localized for associations between trait anxiety and P-selective P1 (Experiment 4; display threshold $p < .005$ uncorrected)

“3). Third, the current evidence remains purely correlational. There is no causal evidence (e.g., TMS or lesion studies) to confirm whether V1 hyperactivity is necessary for anxiety. Additional TMS evidence targeting V1/V2 stimulation could substantially strengthen the argument for early sensory deficits and regional specificity.”

The Reviewer raises a very valid point, one that pertains to a substantial body of cognitive neuroscience literature. As noted in Point 1 above, the stable, enduring nature of personality traits presents inherent challenges for experimental manipulation, making correlational analyses the primary methodology. While causal inference of personality effects ideally requires longitudinal designs, we hope the current study can stimulate such research by drawing attention to an underappreciated association between trait anxiety and visual cortical hyperactivity.

We fully acknowledge the value of experimental manipulations such as TMS of the V1/V2. However, as alluded to in Point 1, such procedures tend to induce relatively transient, state-level shifts. In contrast, our goal here is to isolate enduring, trait-like association, which requires careful control of state-related changes. We have now discussed this important limitation in the revised manuscript.

p. 13: Several limitations to our studies warrant discussion. Trait research, with its focus on stable, enduring predispositions, inherently limits experimental manipulation and relies primarily on correlational analyses. While interventional techniques, such as pharmacological manipulation and neurostimulation, can alter anxiety or sensory processing, they typically induce transient, state-level shifts that might confound the interpretation of trait-level associations. To establish causality, longitudinal designs would be ideal, particularly prospective studies that elucidate a link between trait-like sensory cortical hyperactivity and clinical conditions such as PTSD or autism. Our hope is that the present study by drawing attention to the underrecognized association between trait anxiety and visual cortical hyperactivity, will stimulate such efforts.

“4). Fourth, the M-pathway hypoactivity observed in Experiment 1 was not clear in Experiments 2-4, raising questions about its robustness. The reciprocal P/M relationship (Fig. S1) might reflect task design (e.g., rapid alternation suppressing M responses) rather than being a genuine trait anxiety effect.”

The Reviewer is absolutely correct. Indeed, the M-pathway hypoactivity was not replicated in **Experiments 2-4**. Accordingly, we refrained from drawing any conclusion about this effect and instead focused on the consistently replicated P-pathway hyperactivity. Moreover, prompted by the reciprocal P/M relationship (**Fig. S1**), we designed **Experiments 2&3** to rule out the potential confound in our task design—“rapid alternation suppressing M responses.” We apologize for lack of clarity and have made these points more explicit in the resubmission.

p. 11: Notably, although **Experiment 1** showed magnocellular hypoactivity in trait anxiety, this effect was absent in **Experiments 2 and 3**, where task design (particularly, the parafoveal magnocellular presentation in **Experiment 2**) minimized suppression of magnocellular responses from rapid parvocellular stimulus repetition.

“5). Finally, while trait anxiety represents a risk factor for disorders (e.g., PTSD, autism), the study doesn't examine clinical populations or establish causal links to symptoms. It remains unclear whether anxiety leads to hypersensitivity in sensory regions or whether sensory hypersensitivity contributes to anxiety development.”

The Reviewer again raises a very important point. We fully agree that our correlational analyses cannot establish causality, and prospective, longitudinal studies are needed to address this critical issue. Nonetheless, by demonstrating a robust and replicable association between trait anxiety and sensory hypersensitivity, we trust the current study will catalyze this important and urgently needed line of research. In the resubmission, we have explicitly acknowledged this limitation and called for prospective studies linking this visual cortical hyperactivity mechanistically to clinical disorders.

p. 13: To establish causality, longitudinal designs would be ideal, particularly, prospective studies that elucidate a link between trait-like sensory cortical hyperactivity and clinical conditions such as PTSD or autism. We hope the present study can stimulate such efforts by drawing attention to the underrecognized association between trait anxiety and visual cortical hyperactivity.

Responses to Reviewer 3:

We are encouraged the Reviewer considers the key idea of our study “*very interesting*” and would “*fully agree with it*”. We also greatly appreciate the Reviewer’s thoughtful comments and useful suggestions, including kind references, which have been fully addressed and incorporated in the revised manuscript. We believe the manuscript is substantially improved as a result. Our responses are detailed below (reviewer’s original comments appear in grey shade). Modifications are highlighted in the text and reproduced here in corresponding responses (highlighted in blue).

“The paper by Wu and colleagues describes a set of experiments, two new and two previously published from a different perspective/scope, that reveal trait anxiety-induced hyperactivity in the visual cortex. This hyperactivity manifests very early, starting at 46 ms in one of the experiments. The authors interpret these results as showing that anxiety involves mechanisms that are not mediated in a top-down fashion, but occur in the “earliest processing phase”, before those top-down processes take place. The idea that structures higher in the processing hierarchy (amygdala,

hippocampus, limbic system or prefrontal regions, as mentioned by authors) are not necessarily involved in all neural mechanisms underlying trait anxiety is very interesting and I fully agree with it. However, this article presents several major problems, in my opinion.

1). On the exclusive allocation of effects to the visual cortex and its status as the “earliest processing stage”.

One of the key conclusions of the paper is that visual cortex activity is affected by trait anxiety, and that this cortical region is solely responsible for the observed effects. In relation to this, and throughout the paper, the idea that visual processing begins in the visual cortex is consistently defended. For example, at several (relevant) passages of the paper, such as the abstract or the conclusion paragraph, the authors indicate that the visual cortex is the “earliest processing stage” or “the earliest staging post for external inputs”.

However, every pre-cortical key structure within the visual ascending route, including the retinal ganglion cells and the visual thalamus, are active visual processors. For example, retinal ganglion cells change their response to the very same stimulus depending on the top-down modulation they receive (Warwick et al., 2024). Similarly, the response of the visual thalamus (e.g., the lateral geniculate nucleus -LGN-, which is the main upstream modulator of striate visual cortex) to identical stimuli also varies as a function of task demands (e.g., see reviews by Ghodrati et al., 2017; Halassa and Kastner, 2017; Saalmann and Kastner, 2011; Weyand, 2016).

Clearly, these structures, which are on the main input way to the visual cortex, strongly influence its activity. In other words, the effects detected in the visual cortex could be at least partially due to these precortical stages (whose activity cannot be detected directly by ERPs, but indirectly through their effects on the visual cortex). Importantly, this precortical activity may underlie certain types of anxiety (Milosavljevic et al., 2016), or be affected by anxiety or arousal (e.g., Salay, 2021; Schröder et al., 2020). In the same vein, and crucially, another key finding of the paper is the differential effect of trait anxiety as a function of the two visual streams, parvocellular and magnocellular, which originate in the LGN of the thalamus and reach, without intermediate modulation, the visual cortex at approximately the latency at which the first anxiety effects are reported here.”

We are deeply grateful for these highly insightful and constructive comments. We regret that our focus on cortical processing inadvertently downplayed the potential contribution of low-level (including first-order) visual biases associated with trait anxiety. In the resubmission, we have addressed this omission by incorporating the referenced findings and acknowledging the possibility that flexible visual encoding even at the level of the retina and thalamus may contribute to the VEP biases observed in our study.

p. 14: Finally, our study focused solely on VEPs, which primarily reflect cortical activity. However, recent animal studies show flexible, emotion-related processing in low-level visual areas, including retinal ganglion cells (79-81) and the visual thalamus (82, 83), suggesting that pre-cortical areas may contribute to the early cortical biases observed in trait anxiety.

We have further removed language suggesting involvement of the earliest stage in the entire visual processing stream.

p. 14: In summary, we provide novel empirical evidence for a direct connection between anxiety and sensory traits, arising at the beginning of cortical processing in primary and secondary sensory cortex. That the brain instantiates adaptable sensory regulation at an upstream, low-order post for external inputs has considerable relevance in relation to mental health.

“2). On parvo- and magnocellular activity discrimination based on spatial frequency or luminance/color manipulation.

Please note that manipulation of spatial frequency, and/or luminance and color, does not guarantee the distinction of magno- and parvo-cellular visual processing, as assumed in the paper (although this strategy has often been employed). For example, see these works by Bernt Skottun: Skottun, B. C. (2015). On the use of spatial frequency to isolate contributions from the magnocellular and parvocellular systems and the dorsal and ventral cortical streams. *Neuroscience & Biobehavioral Reviews*, 56, 266-275.

Skottun, B. C. (2013). On using isoluminant stimuli to separate magno-and parvocellular responses in psychophysical experiments—A few words of caution. *Behavior Research Methods*, 45(3), 637-645.”

We again greatly appreciate the Reviewer’s pertinent comments and helpful references. As the Reviewer may agree, non-invasive experimental dissociation of visual pathways in humans remains a significant challenge. Skottun’s reviews were particularly instructive in guiding our efforts to separate magnocellular and parvocellular pathways. Accordingly, our lab has developed a rigorous approach to isolate their respective activation: rather than relying on a single physical property, we designed our laboratory stimuli (e.g., Gabor patches and gratings used in **Experiments 1–3**) by jointly manipulating luminance contrast, color, and spatial frequency in order to maximize pathway segregation.

We acknowledge that the outlined approach does not achieve complete separation. Nonetheless, the distinct VEPs consistently elicited by these stimuli, across distinct experiments, provides support for sufficient pathway dissociation. In the resubmission, we explicitly acknowledge limitations of our approach while pointing to reliable and reproducible VEP profiles observed in our data, which align well with previous findings in the literature.

p. 11: We acknowledge that non-invasive approaches, including stimulus manipulation, cannot achieve a complete visual pathway separation. Nonetheless, our integrated calibration of stimulus luminance, contrast, color, and spatial frequency provides for a maximization of pathway dissociation with consistent elicitation of VEPs characteristic of each pathway.

“3). On the experimental design and logic.

Experiments 2 and 3 were designed de novo for this study (Experiments 1 and 4 consist of

reanalyzing data from past published experiments, but from a new perspective/scope). The authors indicate that Experiment 3 was designed to rule out that the results of Experiment 2 were due to a restricted sensitivity to color processing, so that the stimuli in Experiment 3 were achromatic. However, the authors also indicate that the order of Experiment 2 and 3 was counterbalanced (the participants were the same in both experiments). In other words, the authors designed Experiment 3 at the same time as Experiment 2. The question that arises is why, if the authors considered from the beginning that color could be an interfering factor, they did not go directly to the experiment without color (current Experiment 3).”

We regret confusion arising from the lack of clarity of our reporting. In fact, **Experiment 3** was primarily motivated by **Experiment 1**—specifically, to exclude the alternative explanation that the observed correlation between trait anxiety and C1 magnitudes reflect mere color sensitivity. To make this clear we have now moved this justification to the end of **Experiment 1**.

p. 6: In addition, an alternative explanation to the parvocellular hyperactivity is it reflects a restricted sensitivity for color processing. Based on these considerations and to adjudicate between the aforementioned explanations, we conducted **Experiments 2 and 3**.

“4). On the interpretation of results.

Two of the main interpretations/conclusions contradict (at least apparently) some of the main results. On the one hand, the correlations between anxiety scores and P1 amplitude did not differ as a function of the emotional load of the images (neutral, fearful, and aversive): Experiment 4. However, the authors indicate that the described modulation of visual cortex activity by trait anxiety serves to better detect threat (e.g., “hyperactivity in visual cortex could facilitate overactivation of threat memory, intensifying threat detection and recall,” p. 11). If the anxiety-induced increase in visual cortex activity is similar for threatening and neutral stimuli (Exp. 4), this interpretation does not seem justified.

On the other hand, the key result and conclusion pointing to the parvocellular, but not magnocellular, visual processing stream as responsible for the observed effects are also not clearly in agreement with some interpretation in the Discussion section. For example, “In individuals with high trait anxiety, sensory cortex hyperactivity (and disinhibition) would allow irrelevant or unwanted environmental input to evoke sensory cortical responses”. However, irrelevant or unwanted environmental inputs are typically projected onto non-foveal regions of the retina (in other words, gaze does not focus on them). Considering that retinal projections to P and M layers of the LGN decline to a greater extent in the former case with eccentricity (e.g., Brown et al., 2005), resulting in a magnocellular bias for peripheral vision, this interpretation seems also contradictory.”

We appreciate these thoughtful comments and regret a lack of clarity in our original exposition of our findings. Our discussion focuses on two independent yet interactive mechanisms within sensory cortex: (1) general, non-valanced sensory cortical hyperactivity and (2) long-term

storage of threat/trauma memories. Through the first mechanism, sensory cortex hyperactivity in trait-anxious individuals might lead to excessive sensory output, or “neural noise,” in response to both neutral and threat cues. This general hyperactivity could, in turn, amplify responses to threat cues by engaging the second mechanism: reactivation of threat memory representations stored in the sensory cortex, enhancing threat memory recall. In patients with PTSD, this interaction may contribute to intrusive traumatic memories.

Additionally, we speculate that parvocellular hyperactivity may increase a likelihood that irrelevant/unwanted or subthreshold stimuli—particularly those falling on the fovea—to cause full-blown sensory cortical responses, triggering higher-order processing. We have revised the Discussion in the resubmission to clarify these points.

p. 12: In individuals with high trait anxiety, sensory cortex hyperactivity (and disinhibition) might facilitate processing of irrelevant or unwanted environmental cues, particularly those falling on the fovea, as well as a subthreshold input, to evoke full-blown sensory cortical responses.

p. 12: This general hyperactivity can further interact with another key process in sensory cortex: sensory cortex is recognized as a critical storage site for threat memory (24, 64, 65), especially in individuals with high trait anxiety (25, 26). Consequent to visual cortical hyperactivity, threat memory representations stored in visual cortex could be more likely to be reactivated, resulting in heightened threat memory recall. This type of interaction could, in turn, contribute to intrusive threat or trauma memories commonly seen in anxiety, PTSD, and autism (44, 45, 66).

“5). Other relevant comments.

- Given that the mean amplitude is very dependent on the width of the analyzed window of ERPs (or window of interest -WOI-), please explain how WOI widths were defined.”

The WOI widths were defined according to standard ERP research conventions—centered on the VEP peak latency with widths proportional to the VEP size. These predefined WOIs were further corroborated by our data-driven, point-by-point analyses across the entirety of the observation window (0-300 ms), which isolated intervals where consecutive points showed significant correlations with trait anxiety ($p < .05$), with the entire intervals surviving multiple comparison correction via permutation. The identified intervals encompassed the predefined WOIs. Therefore, the WOIs here are justified both by ERP research conventions and data-driven interval identification.

p. 25: Notably, these lines fully encompassed the VEP windows predefined based on ERP conventions, providing empirical validation for their use.

“- No evidence of effect is not the same as evidence of no effect. The absence of significant correlation between ERP amplitude and anxiety scores is interpreted as a proof in favor of the null hypothesis -i.e., no effect- (e.g., “negating the presence of a magnocellular hypoactivity in trait

anxiety”, p. 7). Indeed, and in order to determine whether this conclusion is valid, the likelihood of H_0 should be computed, for example, through Bayesian methods.”

The Reviewer is absolutely right—we regret the overly strong claims regarding the null effects and have revised the language accordingly in the resubmission. Moreover, we now report the corresponding Bayes factors to provide a more nuanced interpretation of the null results. Briefly, the three experiments consistently showed moderate evidence for absence of relationship between anxiety and magnocellular activity. Furthermore, in a new analysis where we pooled the three independent samples to enhance power, we failed to observe a correlation between trait anxiety and M-selective VEP magnitude, while the correlation between trait anxiety and P-selective VEP was again evident in this analysis.

p. 7: while failing to support the presence of a magnocellular hypoactivity in trait anxiety (see SOM for Bayesian Factors in support of the null finding).

p. 10: For the M-selective VEP, there was no effect of anxiety (\$sr = -.02\$, \$p = .864\$ ), consistent with the main results

SOM p. 6: Bayes Factors for non-significant correlations

To further evaluate the non-significant correlation between trait anxiety and M-selective P1 amplitude, we performed Bayesian analyses to quantify the relative evidence for the null hypotheses. The resulting Bayes factors for **Experiments 2 & 3** were $BF_{10} = 0.177$ and 0.181 , indicating the data were approximately 5.6 and 5.5 times more likely under the null hypothesis than under the alternative. This provides moderate evidence in favor of the absence of a relationship.

SOM p. 7: Bayes Factors for non-significant correlations

For the non-significant correlation between trait anxiety and M-selective P1 amplitude in **Experiment 4**, we performed Bayesian analyses to quantify the relative evidence for the null hypotheses. The resulting Bayes factor was $BF_{10} = 0.299$, indicating that the data were approximately 3.3 times more likely under the null hypothesis than under the alternative. This provides moderate evidence in favor of the absence of a relationship.

“References

- Brown, L. E., Halpert, B. A., & Goodale, M. A. (2005). Peripheral vision for perception and action. *Experimental Brain Research*, 165, 97–106.
- Ghodrati, M., Khaligh-Razavi, S., & Lehky, S. R. (2017). Towards building a more complex view of the lateral geniculate nucleus: Recent advances in understanding its role. *Progress in Neurobiology*, 156, 214-255.
- Halassa, M. M., & Kastner, S. (2017). Thalamic functions in distributed cognitive control. *Nature*

Neuroscience, 20(12), 1669-1679.

Milosavljevic, N., Cehajic-Kapetanovic, J., Procyk, C. A., & Lucas, R. J. (2016). Chemogenetic activation of melanopsin retinal ganglion cells induces signatures of arousal and/or anxiety in mice. *Current Biology*, 26(17), 2358-2363.

Saalmann, Y. B., & Kastner, S. (2011). Cognitive and perceptual functions of the visual thalamus. *Neuron*, 71, 209-223.

Salay, L. D., & Huberman, A. D. (2021). Divergent outputs of the ventral lateral geniculate nucleus mediate visually evoked defensive behaviors. *Cell Reports*, 37(109792).

Schröder, S., Steinmetz, N. A., Krumin, M., Pachitariu, M., Rizzi, M., Lagnado, L., ... & Carandini, M. (2020). Arousal modulates retinal output. *Neuron*, 107(3), 487-495.

Warwick, R. A., Riccitelli, S., Heukamp, A. S., Yaakov, H., Swain, B. P., Ankri, L., ... & Rivlin-Etzion, M. (2024). Top-down modulation of the retinal code via histaminergic neurons of the hypothalamus. *Science Advances*, 10(35), eadk4062."

We are deeply grateful for the excellent references provided by the Reviewer. We have incorporated most of them into the revised manuscript, and they have significantly enriched and deepened both our conceptual thinking and the manuscript itself.

References

- Clark, V. P., & Hillyard, S. A. (1996). Spatial selective attention affects early extrastriate but not striate components of the visual evoked potential. *Journal of cognitive neuroscience*, 8(5), 387-402.
- Foxe, J. J., & Simpson, G. V. (2002). Flow of activation from V1 to frontal cortex in humans: A framework for defining "early" visual processing. *Experimental brain research*, 142, 139-150.
- Hagler, D. J., Jr., Halgren, E., Martinez, A., Huang, M., Hillyard, S. A., & Dale, A. M. (2009). Source estimates for MEG/EEG visual evoked responses constrained by multiple, retinotopically-mapped stimulus locations. *Hum Brain Mapp*, 30(4), 1290-1309. <https://doi.org/10.1002/hbm.20597>
- Krusemark, E. A., & Li, W. (2011). Do all threats work the same way? Divergent effects of fear and disgust on sensory perception and attention. *Journal of Neuroscience*, 31(9), 3429-3434.
- Krusemark, E. A., & Li, W. (2013). From early sensory specialization to later perceptual generalization: dynamic temporal progression in perceiving individual threats. *Journal of Neuroscience*, 33(2), 587-594.
- Li, W., & Keil, A. (2023). Sensing fear: fast and precise threat evaluation in human sensory cortex. *Trends in cognitive sciences*, 27(4), 341-352.
- Li, W., Zinbarg, R. E., Boehm, S. G., & Paller, K. A. (2008). Neural and behavioral evidence for affective priming from unconsciously perceived emotional facial expressions and the influence of trait anxiety. *Journal of cognitive neuroscience*, 20(1), 95-107.
- Li, Z., Yan, A., Guo, K., & Li, W. (2019). Fear-related signals in the primary visual cortex. *Current Biology*, 29(23), 4078-4083. e4072.
- Martinez, A., Anllo-Vento, L., Sereno, M. I., Frank, L. R., Buxton, R. B., Dubowitz, D., Wong, E. C., Hinrichs, H., Heinze, H. J., & Hillyard, S. A. (1999). Involvement of striate and extrastriate visual cortical areas in spatial attention. *Nature Neuroscience*, 2(4), 364-369.
- Ossenblok, P., & Spekreijse, H. (1991). The extrastriate generators of the EP to checkerboard onset: A source localization approach. *Electroencephalography & Clinical Neurophysiology: Evoked Potentials*, 80(3), 181-193. [https://doi.org/10.1016/0168-5597\(91\)90120-M](https://doi.org/10.1016/0168-5597(91)90120-M)
- You, Y., Brown, J., & Li, W. (2021). Human Sensory Cortex Contributes to the Long-Term Storage of Aversive Conditioning. *J Neurosci*, 41(14), 3222-3233. <https://doi.org/10.1523/JNEUROSCI.2325-20.2021>